# APD: Boosting Adversarial Transferability via Perturbation Dropout

## Abstract

The transferability of adversarial attack to deep neural networks (DNNs) accounts for the possibility that the adversarial examples crafted for a known model can also mislead other unseen models in black-box setting. Existing literature to improve the adversarial transferability often focus on spreading the adversarial perturbations towards the whole image, which can be counter-productive as the extended perturbation can hardly track the attention regions across different models. That's because although they spread the perturbation throughout the entire image but they do not consider the mutual influence of different perturbation regions. In this paper, we propose a simple yet effective perturbation-dropping scheme that can enhance the transferability of the adversarial examples by incorporating the dropout mechanism during their optimization process. Specifically, we leverage the class activation map (CAM) to locate the midpoint of the dropped regions, whereby the effective perturbation can be generated for the target models while maintaining the attack rate towards the source model even if some blocks of the perturbation noises are dropped. Extensive experiments are conducted on the ImageNet dataset, which demonstrates that the proposed method outperforms state-of-the-art methods, that achieve both high attack efficiency and transferability.

## 1 Introduction

In recent years, Deep Neural Networks (DNNs) have greatly reshaped modern people's lives via various intelligent applications, such as face recognition, autonomous driving, etc. Despite the ever-growing applications, DNNs are known to be vulnerable to adversarial attacks, which can let a model misclassify a certain data sample that is added with a carefully crafted perturbation either in white-box (model is known to attacker) or black-box settings (model is unknown).

We specifically focus on adversarial attacks in the black-box setting which is common in practical scenarios. The basic idea of black-box adversarial attacks is to craft adversarial examples using the source model known to the attackers and then deceive the unknown target model. The principle behind this idea is that the impact of an attack on the source model can be extended to a target model, which is termed as the transferability of the adversarial sample. Such adversarial transferability has garnered considerable attention lately to exploit the working mechanisms behide it, such as in Wu et al. (2018); Liu et al. (2016); Tramèr et al. (2017b).

To improve the success rate of black-box adversarial attacks, many approaches have recently been proposed to enhance adversarial transferability, e.g., Xie et al. (2019b); Dong et al. (2019); Lin et al. (2019); Wang et al. (2021); Huang et al. (2019); Qin et al. (2022); Li et al. (2020). As shown in Figure 1, some studies Dong et al. (2019); Gao et al. (2020) show that different models may have distinct attention regions when recognizing objects. The perturbation region of adversarial examples generated based on a source model is highly correlated with the source model's attention region, which may be inconsistent with the attention region of the target model, leading to dissatisfactory transferability. Considering this, they suggested extending the perturbations generated by the source model to cover the entire image or object region, thereby increasing the transferability of perturbations across models.

However, such extending strategies usually craft perturbations by taking them as a whole, and thus causing the attack effect to diminish. Specifically, they update perturbations by iteratively computing gradients across all regions of the image together during the attack optimization. Therefore,

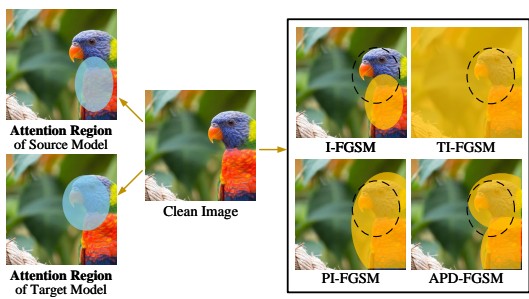 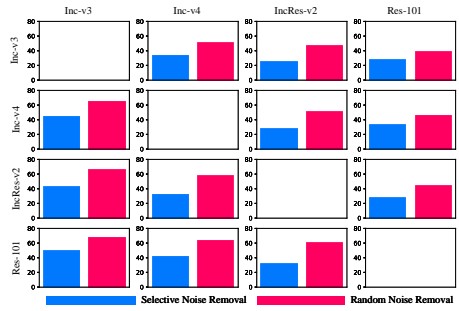

(a) Comparison of different attack methods        (b) Impact of different drop strategies

Figure 1: 1(a) Illustration of attention regions of different models and perturbation regions of different attack methods. **Left:** The attention region (class activation map Chattopadhay et al. (2018)) of both the source model and the target model. **Right:** Adversarial examples are generated on the source model using four different methods, where the perturbation regions are in orange and the dashed circle denotes the attention region of the target model. The perturbation region of I-FGSM Kurakin et al. (2018) is highly related to the attention region of the source model. TI-FGSM Dong et al. (2019) extends the perturbation to cover the whole image. PI-FGSM Gao et al. (2020) surrounds the object with perturbations. Our APD-FGSM generates independent perturbations regions. 1(b) We conducted multiple experiments using various source models and target models. The leftmost column of the images represents the source models, while the top row represents the target model. Results show Selective Noise Removal can induce a more significant decrease in the attack success rate.

when attacking deep neural networks, the perturbations across different regions need to work synergistically and thus the adversarial attack is most effective only when all the perturbations in different regions can take effect jointly. However, considering the fact that most models only focus on limited attention regions, the attack effect of the extending strategies will be greatly limited due to the lack of assistance with other region's perturbations. As illustrated in Figure 1(a), methods like TI-FGSM Dong et al. (2019) and PI-FGSM Gao et al. (2020) extend perturbations across the whole image or object. But the target model can only focus on perturbations within its attention area. We can divide perturbations into two categories: attention perturbations that the model focuses on, and neglected perturbations outside its attention. During the attack phase, attention perturbations play the primary role. Without the synergy from neglected perturbations, attacks may fail despite extending perturbations across the image. To further our exploration, we conduct experiments to reveal the synergy from neglected perturbations. We test it by comparing two distinct noise dropping approaches: 1).Selective Noise Removal: In this experiment, we remove the noise in which the source model focuses while the target model does not. 2). Random Noise Removal: For comparison, we randomly drop noise of equal size as in the Selective Noise Removal. The results are shown in Figure 1(b), which reveals the critical role of synergy from neglected perturbations.

Considering the limitations of existing works, we are inspired to explore whether we could further enhance transferability by reducing the synergy(i.e. mutual influence) between different perturbations, which would enable successful attacks even when part of the perturbations is lost. To achieve this goal, we propose the Adversarial Perturbation Dropout attack method (APD), which breaks the synergy of perturbations to improve the transferability of the adversarial sample across different models. Our method is motivated by the neuron dropout for training DNNs Srivastava et al. (2014), whereby the model can be viewed as the ensemble from multiple sub-models due to the decoupling effect of dropout, and is demonstrated to achieve better model generalization. In our solution, different from traditional model parameters dropout Srivastava et al. (2014), we adopt the dropout mechanism to the original data sample instead of the model.

Specifically, we remove some perturbations of the sample at each round and constitute a mini-batch for updating the adversarial perturbations. This mini-batch of samples consists of different dropped versions of the original sample, which can significantly stabilize the training process. To focus on the attention region, we also leverage class activation map (CAM) to determine the midpoints of the dropped areas to achieve better attack performance and we will give the reason later. Additionally,

we generate input examples by ensembling different sizes of dropped regions. As shown in Figure 1, our APD method's perturbations are clearly divided into two independent parts, whereas the perturbations of the other three methods are tangled with each other.

We conducted extensive experiments on the ImageNet dataset Russakovsky et al. (2015) to demonstrate the effectiveness of our method in improving attack success rates in black-box settings. We found that our APD method can be seamlessly integrated with other adversarial attack methods to achieve even higher attack success rates. The experimental results show that our proposed attack method outperforms state-of-the-art methods by a significant margin. Overall, these results demonstrate the effectiveness of our approach and highlight its potential for use in real-world applications. Our contributions can be summarized as follows.

- We identify that the synergy of perturbations may reduce the transferability of the adversarial example. To tackle this challenge, we propose utilizing perturbation dropout to increase the independence of perturbations across different attention regions such that the target model can still be successfully attacked, even if it only pays attention to partial perturbations.

- To improve the effectiveness of perturbation dropout, we propose leveraging class attention maps to dropout perturbations. The main idea of using CAM is to identify the main attention regions such that the synergy of perturbations can be broken thoroughly.

- We carry out extensive experiments on various datasets and different settings. Experimental results illustrate that our proposed method significantly improves the attack success rates as combined with existing methods, e.g., by up to 19.6%.

In the following text, related works are first introduced in Section 2. Then, the proposed methods are specified in Section 3. Section 4 presents the evaluations of the proposed method. Finally, Section 5 concludes this paper.

## 2 RELATED WORKS

### 2.1 ADVERSARIAL EXAMPLES

Adversarial examples are typically created by adding small perturbations that are imperceptible to humans to clean inputs. Since the introduction of the Fast Gradient Sign Method (FGSM)Goodfellow et al. (2014) for generating adversarial examples, many other adversarial attack methods have been proposed, such as Projected Gradient Descent (PGD)Madry et al. (2017), Iterative-Fast Gradient Sign Method (I-FGSM)Kurakin et al. (2018), and Carlini and Wagner (C&W)Carlini & Wagner (2017) attack. Black-box attacks are in contrast to white-box attacks, where the attacker has complete knowledge of the target model, including its architecture and parameters, and can directly exploit this information to generate highly targeted adversarial examples. Although it is challenging, many methods have been proposed to enhance the transferability of adversarial examples, including techniques like Xie et al. (2019b); Dong et al. (2019); Wang et al. (2021); Lin et al. (2019); Wu et al. (2018); Huang et al. (2019); Qin et al. (2022); Dong et al. (2018). These methods seek to improve the attack success rate in black-box settings. For the purposes of our paper, we focus solely on the task of image recognition. Our aim is to explore new methods for generating adversarial examples in the context of image recognition.

### 2.2 DEFENSE TO ADVERSARIAL EXAMPLES

Currently, there are numerous defense methods available against adversarial attacks. Adversarial training Goodfellow et al. (2014) is a well-known technique that aims to improve the robustness of deep neural network models against adversarial examples by training the model using both clean and adversarial examples. Ensemble adversarial training Tramèr et al. (2017a) proposes using the perturbations of other pre-trained models to increase the diversity of adversarial examples during adversarial training, further enhancing the model's robustness. Feature denoising Xie et al. (2019a) is another approach that filters out irrelevant or noisy features from input data before feeding it into the model. This helps to reduce the susceptibility of the model to attacks designed to exploit such features. NRP Naseer et al. (2020) first filters out the noise for adversarial attacks before inputting

them into the model. In addition to defending against adversarial attacks, black-box attack methods aim to increase the success rate of attacking these models.

## 3 METHODS

In this section, we first define the notations in this paper. Then, we briefly summarize existing methods regarding the transferability of adversarial attack. After that we present the detailed design of APD.

### 3.1 NOTATION

Let $x$ denote a benign image and $x^{adv}$ denote the corresponding adversarial example. Let $y^{true}$ represent the ground truth label of $x$. Let $f(x)$ be the classifier and $J(x, y)$ represent the loss function of classifier $f$. The goal of adversarial attack is to maximize $J(x^{adv}, y^{true})$ to mislead the model's prediction. We require that the adversarial example $x^{adv}$ satisfies $\|x - x^{adv}\|_p < \epsilon$ simultaneously, where $\epsilon$ is the maximum adversarial perturbation allowed. In this work, we focus on $p = \infty$ as in previous works, i.e., the maximum perturbation in each pixel is constrained by $\epsilon$.

### 3.2 EXISTING METHODS

The existing methods about adversarial examples and transferability are listed as follows.

**I-FGSM:** Kurakin et al. (2018) propose an iteration based fast gradient sign method to generate adversarial example:

$$x_0^{adv} = x, \quad x_{t+1}^{adv} = Clip_x^{\epsilon} \left\{ x_t^{adv} + \alpha \cdot sign \left( \nabla_{x_t^{adv}} J \left( x_t^{adv}, y^{true} \right) \right) \right\},$$

where $Clip_x^{\epsilon} (\cdot)$ function constrain the input between $x - \epsilon$, $x + \epsilon$ by making it equal to $x + \epsilon$ if it is greater than $x + \epsilon$, or equal to $x - \epsilon$ if it is less than $x - \epsilon$.

**MI-FGSM:** Dong et al. (2018) proposed a novel technique to improve the efficiency and effectiveness of adversarial attacks called momentum iterative fast gradient sign method (MI-FGSM). MI-FGSM is an extension of the fast gradient sign method (FGSM) that introduces momentum to the original gradient. The momentum term is used to track the direction of the previous iteration's gradient and amplify it to generate more effective adversarial examples, which is defined as follows.

$$g_{t+1} = \mu \cdot g_t + \frac{\nabla_{x_t^{adv}} J \left( x_t^{adv}, y^{true} \right)}{\|\nabla_{x_t^{adv}} J \left( x_t^{adv}, y^{true} \right)\|_1}, \quad x_{t+1}^{adv} = Clip_x^{\epsilon} \left\{ x_t^{adv} + \alpha \cdot sign \left( g_{t+1} \right) \right\}, \quad (1)$$

where $\mu$ is the decay factor.

### 3.3 EXPLORING MORE EFFECTIVE DROP TECHNIQUE

Before introducing our method detailedly, we first explain how to select regions to drop perturbations. There is a simple and straightforward method that come to mind, i.e. randomly choosing some patches in the image to drop. However, like guided mask in Wu et al. (2022), we believe random choice is also not an effective strategy. To find more effective way, we need to explore which synergies between perturbation regions in an image have greater impact on transferability across models.

We first argue that the attention region of a model is composed of a limited number of blocks. By observing the CAMs of several models on an certain adversarial example, we found that attention regions often locate in a few particular blocks. As shown in Figure 2, when a DNN infer the category of an image of loggerhead turtle, some 'hot regions', e.g., the turtle's head, legs, shell, are focused, i.e., paid with much attention to distinguish from other categories. As stated above, the attention region of a model is composed of a limited number of blocks. This fact, along with the reality that different models have different attention blocks, inspired us that the synergies between perturbations in these blocks may have greater impact on transferability across models. In A.1, we verify our thought empirically.

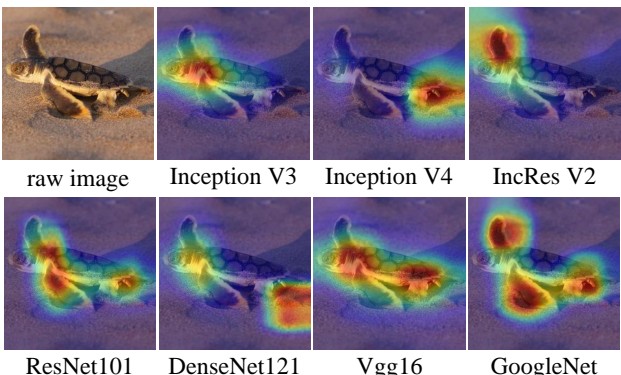

raw image    Inception V3    Inception V4    IncRes V2

ResNet101    DenseNet121    Vgg16    GoogleNet

Figure 2: We utilized several models to generate adversarial examples and examined their class activation maps. We employed a variety of models to increase the diversity of the attention regions, including Inception V3 Szegedy et al. (2016), Inception V4 Szegedy et al. (2017), Inception-Resnet-V2 Szegedy et al. (2017), ResNet101 He et al. (2016), DenseNet121 Huang et al. (2017), Vgg16 Simonyan & Zisserman (2014) and GoogleNet Szegedy et al. (2015). We can observe that the attention regions of different models are comprised of a limited number of blocks. For example, in GoogleNet, the attention region is a combination of the left and right forelegs and hind legs. For Inception V3, it focuses solely on the turtle's head.

## 3.4 ADVERSARIAL PERTURBATION DROPOUT ATTACK

As stated by Lin et al. (2019), the process of generating adversarial examples can be analogous to training a deep neural network model. Therefore, improving the transferability of adversarial examples can apply similar scheme for improving the generalization of a model. To achieve this, we employ dropout Srivastava et al. (2014), a commonly used yet simple method to improve the generalization of deep models, to improve the transferability of adversarial examples.

Briefly speaking, we integrate the dropout mechanism into the I-FGSM Kurakin et al. (2018) to enhance transferability. At each step of I-FGSM, we first select a set of square image regions. Perturbations in these regions are then dropped one-by-one to generate new images - dropping each region produces a separate image. For example, if we want to choose three regions, then dropping perturbations at each of them would produce three images. These dropout-modified images are optimized in the current step and averaged afterwards. This process repeats for a fixed number of steps, resulting in the final adversarial example.

As stated in 3.3, we propose using CAM-guided dropping method in our attack. Specifically, we use hotspots(i.e. the local maximum points) of the CAM as the midpoints of the dropped regions. We provide an example in Figure 3 to illustrate how to obtain dropped blocks from the CAM. It's worth noting that we use CAMs at each attack iteration instead of just the initial image's CAM because the attention region expands over the attack steps. The initial image's CAM provides limited attention blocks. Using CAMs at each step allows us to identify as many attention blocks as possible over the complete attack sequence, providing a more comprehensive set of regions to drop for better transferability.

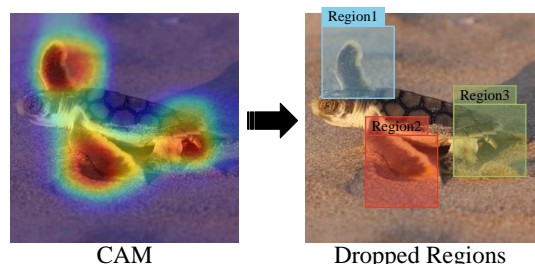

CAM        Dropped Regions

Figure 3: An example illustrating how to determine dropped regions by the CAM.

We apply CAM to identify the main attention regions, dropping the corresponding blocks separately has the potential to reduce the synergy of these regions, and thus improve the attack transferability for target model. Besides, applying random selection may lead to dropping out useless blocks(e.g. background) that have no contribution to the model. Also, if a random selected block contains

multiple attention hotspots, then dropping out the whole block would not break the synergy of these hotspots, and thus could not improve transferability.

There is another issue to consider, i.e., how to determine the size of each block. To address this, we adopt an ensemble of square regions with varied size to conduct the attack.

As discussed above all, we propose the Adversarial Perturbation Dropout (APD) attack method to enhance transferability. Here, we provide a detailed description of our proposed method. At each iteration, we start by generating the CAM on the current version of the image $x_t^{adv}$, using Grad-CAM++ Chattopadhay et al. (2018) as our CAM generation function. Then we take the local maximum points as midpoints of the dropped region. Let $n$ be the number of midpoints and we limit it to 3. Afterwards, for each mid-point, we generate $m$ images by dropping a square region with a side length taken from the set $\{\beta, 2\beta, ..., m\beta\}$, where $\beta$ is the scaling factor. We feed these $nm$ images to our source model to get a set of gradients. Finally, we calculate the mean of gradients to update $x_t^{adv}$. In summary, our optimization procedure (without momentum) is as follows:

$$ x_0^{adv} = x, \quad x_{t+1}^{adv} = Clip_x^\epsilon \left\{ x_t^{adv} + \alpha \cdot sign \left( \frac{1}{nm} \cdot \sum_{j=1}^{n} \sum_{k=1}^{m} \nabla_{x_{tjk}^{drop}} J \left( x_{tjk}^{drop}, y^{true} \right) \right) \right\} $$

, where $x_t^{adv}$ is the adversarial example at the $t$-th iteration. $x_{tjk}^{drop}$ is the image generated by dropping the perturbations located at the $j$-th center point with scale $k\beta$ from $x_t^{adv}$. $Clip_x^\epsilon (\cdot)$ function constrain the input between $x - \epsilon$, $x + \epsilon$ by making it equal to $x + \epsilon$ if it is greater than $x + \epsilon$, or equal to $x - \epsilon$ if it is less than $x - \epsilon$. The detailed method is described in Algorithm 1 in A.4 of *Appendix*.

## 4 EXPERIMENTS

In this section, we present experimental results on the widely-used ImageNet dataset Russakovsky et al. (2015) to show the effectiveness of our proposed methods, including the experiment settings, evaluate results and the robustness.

### 4.1 EXPERIMENTAL SETUP

**Dataset and Models.** We conduct experiments on ImageNet dataset Russakovsky et al. (2015). We randomly choose 1000 images from 1000 categories. The chosen images can be almost classified correctly by all models, i.e., the model has a high accuracy. We adopt four normally trained models, i.e. Inception-v3 (Inc-v3) Szegedy et al. (2016), Inception-v4 (Inc-v4)Szegedy et al. (2017), Inception-Resnet-v2 (IncRes-v2) Szegedy et al. (2017) and Resnet-v2-101 (Res-101) He et al. (2016) and three adversarially trained models, i.e. Inc-v3$_{ens3}$, Inc-v3$_{ens4}$ and IncRes-v2$_{ens}$ Tramèr et al. (2017a).

**Baselines and Implementation Details.** We use the MI-FGSM Dong et al. (2018) as the baseline. In addition, we also use some input transformations method that are integrated into MI-FGSM as our baselines including DIM Xie et al. (2019b), TIM Dong et al. (2019), SIM Lin et al. (2019) and AAM Wang et al. (2021). Besides, we also consider the state-of-the-art input transformations combination method, i.e. AA-TI-DIM Wang et al. (2021). We take the same settings as Dong et al. (2018) with the maximum perturbation of $\epsilon = 16$, number of iteration $T = 10$, step size $\alpha = 1.6$. For MI-FGSM, we set the decay factor for MI-FGSM $\mu = 1.0$. For DIM, we set the transformation probability to 0.5. For TIM, we use Gaussian kernel and set its kernel size to $7 \times 7$. For SIM, we set the number of scale copies $m1 = 5$ and scale factor $\gamma_i = 1/2^i$. For AAM, we set randomly sample number $m_2 = 3$ and $\eta = 0.2$. And we set $m = 5$ and $\beta = 27$ for our method.

### 4.2 THE EVALUATION OF GENERAL ATTACKS

**Single Model Attack.** We first demonstrate the performance of our methods by attacking with a single model. We choose the MI, DIM, TIM, SIM, and AAM as baselines. We integrate our method into all the baselines, such as we let 'MI/+APD' denote MI and APD-MI, respectively. The criteria is attack success rate (ASR) which means that the misclassification rates by target models. We

Table 1: **Attack Success Rates (%) of adversarial attacks against four normally trained models and three ensemble adversarially trained models under single model setting.** We integrate our method into all the baselines, such as we let 'MI/+APD' denote MI and APD-MI. The best results are bold. * indicates white-box setting.

a Crafted on Inc-v3.

| Attack | Inc-v3* | Inc-v4 | IncRes-v2 | Res-101 | Inc-v3$_{ens3}$ | Inc-v3$_{ens4}$ | IncRes-v2$_{ens}$ |
|---|---|---|---|---|---|---|---|
| MI/+APD | **100.0/100.0** | 50.0/**67.2** | 47.8/**67.4** | 38.1/**57.5** | 14.4/**22.7** | 12.7/**21.7** | 6.2/**9.5** |
| DIM/+APD | **100.0/100.0** | 69.8/**85.9** | 66.0/**84.8** | 51.5/**75.4** | 17.3/**27.2** | 15.4/**25.7** | 7.1/**12.7** |
| TIM/+APD | **100.0/100.0** | 58.1/**74.3** | 56.2/**70.5** | 45.1/**62.1** | 17.1/**24.8** | 16.1/**24.9** | 7.5/**12.4** |
| SIM/+APD | **100.0/100.0** | 79.7/**90.4** | 76.8/**89.5** | 71.5/**88.5** | 27.4/**35.1** | 25.0/**33.8** | 13.7/**15.9** |
| AAM/+APD | **100.0/100.0** | 82.5/**95.2** | 79.4/**92.8** | 76.3/**90.7** | 29.3/**35.5** | 25.2/**35.3** | 13.1/**17.0** |
| AA-TI-DIM/+APD | **100.0/100.0** | 90.6/**96.7** | 87.6/**95.1** | 83.4/**91.9** | 47.7/**50.5** | 41.9/**47.6** | 24.4/**26.6** |

b Crafted on Inc-v4.

| Attack | Inc-v3 | Inc-v4* | IncRes-v2 | Res-101 | Inc-v3$_{ens3}$ | Inc-v3$_{ens4}$ | IncRes-v2$_{ens}$ |
|---|---|---|---|---|---|---|---|
| MI/+APD | 67.2/**80.7** | **100.0/100.0** | 50.5/**70.7** | 45.4/**65.2** | 14.3/**24.6** | 12.9/**22.3** | 6.5/**10.2** |
| DIM/+APD | 79.2/**92.3** | **100.0/100.0** | 68.2/**87.0** | 57.7/**79.8** | 16.9/**29.1** | 15.5/**28.3** | 8.2/**14.0** |
| TIM/+APD | 71.9/**85.3** | 99.9/**100.0** | 59.5/**75.7** | 49.7/**71.0** | 17.5/**28.2** | 17.6/**29.4** | 8.8/**15.5** |
| SIM/+APD | 86.1/**96.3** | **100.0/100.0** | 81.5/**93.4** | 77.7/**93.0** | 33.3/**43.5** | 29.1/**39.5** | 17.3/**20.7** |
| AAM/+APD | 88.2/**97.7** | **100.0/100.0** | 82.4/**96.8** | 77.8/**92.6** | 34.9/**53.1** | 30.0/**44.7** | 16.1/**23.9** |
| AA-TI-DIM/+APD | 91.8/**97.6** | **100.0/100.0** | 88.0/**96.2** | 81.5/**89.5** | 51.0/**53.3** | 48.5/**53.3** | 31.3/**34.9** |

c Crafted on IncRes-v2.

| Attack | Inc-v3 | Inc-v4 | IncRes-v2* | Res-101 | Inc-v3$_{ens3}$ | Inc-v3$_{ens4}$ | IncRes-v2$_{ens}$ |
|---|---|---|---|---|---|---|---|
| MI/+APD | 66.2/**82.7** | 57.3/**77.1** | **99.9**/99.8 | 46.1/**67.5** | 14.2/**25.9** | 14.8/**24.7** | 7.9/**14.8** |
| DIM/+APD | 83.9/**94.1** | 79.5/**90.0** | 99.5/**99.9** | 64.0/**83.7** | 17.7/**33.7** | 16.8/**32.8** | 9.7/**17.9** |
| TIM/+APD | 73.6/**86.6** | 64.2/**82.6** | 99.8/**99.9** | 53.7/**74.0** | 16.7/**30.9** | 16.9/**28.1** | 10.4/**17.3** |
| SIM/+APD | 87.8/**97.7** | 84.9/**95.8** | **100.0/100.0** | 79.3/**94.0** | 36.9/**49.0** | 30.4/**41.3** | 20.6/**27.9** |
| AAM/+APD | 90.8/**98.3** | 86.5/**95.8** | **100.0/100.0** | 81.8/**95.8** | 39.5/**53.1** | 31.6/**44.1** | 21.3/**30.3** |
| AA-TI-DIM/+APD | 95.3/**99.0** | 94.2/**98.6** | **100.0/100.0** | 91.1/**97.5** | 61.8/**74.5** | 52.7/**62.1** | 38.9/**50.2** |

d Crafted on Res-101.

| Attack | Inc-v3 | Inc-v4 | IncRes-v2 | Res-101* | Inc-v3$_{ens3}$ | Inc-v3$_{ens4}$ | IncRes-v2$_{ens}$ |
|---|---|---|---|---|---|---|---|
| MI/+APD | 67.6/**79.0** | 61.8/**72.8** | 59.8/**73.1** | **99.9**/99.6 | 18.6/**28.6** | 15.8/**27.8** | 8.5/**14.3** |
| DIM/+APD | 88.1/**96.5** | 85.0/**92.7** | 83.1/**92.9** | 99.7/**99.7** | 26.0/**37.4** | 22.8/**34.4** | 11.9/**17.7** |
| TIM/+APD | 72.6/**83.7** | 63.6/**75.5** | 66.7/**77.2** | 99.6/**99.6** | 23.3/**34.0** | 23.0/**32.3** | 12.6/**20.1** |
| SIM/+APD | 77.8/**88.4** | 75.2/**85.7** | 75.3/**87.2** | 99.9/**100.0** | 33.3/**44.8** | 29.9/**41.6** | 18.0/**22.0** |
| AAM/+APD | 82.3/**91.3** | 78.6/**89.4** | 77.6/**89.2** | **100.0/100.0** | 36.4/**47.6** | 32.6/**43.0** | 18.4/**23.8** |
| AA-TI-DIM/+APD | 91.3/**98.3** | 90.5/**97.6** | 91.1/**98.5** | **100.0/100.0** | 66.1/**79.5** | 57.7/**67.2** | 40.7/**52.2** |

use Inc-v3, Inc-v4, IncRes-v2 and Res-101 as source models as well as Inc-v3, Inc-v4, IncRes-v2 and Res-101, Inc-v3$_{ens3}$, Inc-v3$_{ens4}$ and IncRes-v2$_{ens}$ as target models. The results are shown in Table 1.

We can observe that the proposed method outperform the baselines by a large margin on all the black-box models.For average attack success rate of all target models, APD outperforms the MI by 12.7%. By integrating to DIM, TIM, SIM, AAM and AA-TI-DIM, APD gets the improvements by 12.7%, 12.3%, 10.3%, 11.0% and 6.8%, respectively. This convincingly validates the high effectiveness of the proposed method. And we reach the state-of-the-art input transformation method for boosting adversarial transferability. Overall, our method can reach an average improvement of 12.74% for black-box normally trained model and 9.00% in adversarial training model.

**Ensemble Model Attack.** Liu et al. (2016) introduce that attacking a combination of models simultaneously could largely boost the adversarial transferability. We also test our method under the ensemble model setting. Specifically speaking, we attack an ensemble of four normally trained models, i.e. Inc-v3, Inc-v4, IncRes-v2 and Res-101. Following Dong et al. (2018), we fuses the logit outputs of different models to attack target models. The results are shown in Table 2.

The results show that there exists a clear margin between proposed method and all baseline methods. Our attack method significantly boosts the transferability by an average value 15.62% in black-box setting.

## 4.3 THE EVALUATION OF ATTACK ON ADVANCED DEFENSE MODELS AND DIVERSE NETWORK ARCHITECTURES

To further demonstrate the effectiveness of APD, we evaluate our method on some defense method and diverse network architectures. For defense method, we consider two kinds of advanced defense methods including feature denoising (FD) Xie et al. (2019a) and purification defense (NRP) Naseer et al. (2020). For diverse network architectures, we consider three models including sequencer deep lstm(Seq2d_l) Tatsunami & Taki (2022), ViT-B/16 Dosovitskiy et al. (2020) and MnasNet Tan et al. (2019). We use $Res152_D$ and $ResNeXt_{DA}$ to refer to two feature denoising models: ResNet152 Denoise and ResNeXt101 DenoiseAll, respectively. Since NRP is a module to remove perturbations of adversarial examples, we adopt NRP as a preprocessing module in front of Inc-v3$_{ens3}$. We use NRP and $NRP_{resG}$ to refer to two purification defense models: NRP based on DenseNet and ResNet, respectively.

We compare the AA-TI-DIM Wang et al. (2021) and proposed APD-AA-TI-DIM (i.e., APD integrated with AA-TI-DIM)), and we adopt the ensemble-model attack in Sec. 4.2. The attack success rate on defense models and diverse network architectures are demonstrate in Table 3. For defense methods, the average attack success rate of APD-AA-TI-DIM is 2.6% higher than that of AA-TI-DIM. For diverse network architectures, the average attack success rate of APD-AA-TI-DIM on Seq2d_l, ViT-B/16 and MnasNet is 13.3%, 11.3% and 1.8% higher than that of AA-TI-DIM.

Table 2: **Attack Success Rates (%) of adversarial attacks under ensemble model setting.** The best results are bold. * indicates white-box setting.

| Attack | Inc-v3* | Inc-v4* | IncRes-v2* | Res-101* | Inc-v3$_{ens3}$ | Inc-v3$_{ens4}$ | IncRes-v2$_{ens}$ |
|---|---|---|---|---|---|---|---|
| DIM/+APD | **100.0**/100.0 | **100.0**/100.0 | 99.8/**100.0** | **100.0**/100.0 | 34.5/**53.3** | 31.1/**55.7** | 18.2/**28.4** |
| TIM/+APD | **100.0**/100.0 | **100.0**/100.0 | 99.7/**99.9** | **100.0**/100.0 | 36.9/**56.7** | 35.5/**56.7** | 20.8/**33.1** |
| SIM/+APD | **100.0**/100.0 | 99.9/**100.0** | **100.0**/100.0 | **100.0**/100.0 | 59.5/**72.7** | 49.7/**68.8** | 32.4/**40.7** |
| AAM/+APD | **100.0**/100.0 | 99.9/**100.0** | **100.0**/100.0 | **100.0**/100.0 | 62.6/**75.5** | 53.2/**69.8** | 34.7/**45.2** |
| AA-TI-DIM/+APD | **100.0**/100.0 | **100.0**/100.0 | **100.0**/100.0 | **100.0**/100.0 | 73.7/**87.9** | 64.7/**86.0** | 43.8/**55.3** |

Table 3: **Attack Success Rates (%) of adversarial attacks under four advanced defense models and three diverse network architectures.** The best results are bold.

| Attack | ResNeXt$_{DA}$ | Res152$_D$ | NRP | NRP$_{resG}$ | Seq2d_l | ViT-B/16 | MnasNet |
|---|---|---|---|---|---|---|---|
| AA-TI-DIM | 30.2 | 40.6 | 9.9 | 8.2 | 78.9 | 69.9 | 97.7 |
| APD-AA-TI-DIM | **34.6** | **43.6** | **11.8** | **9.3** | **92.2** | **81.2** | **99.5** |

## 4.4 ABLATION STUDY

**Comparison of Random Selection and CAM-based Selection.** Previously, we proposed using CAMs to locate important regions to drop perturbations. We compare randomly selecting regions to drop versus using CAMs to locate drop regions. Random selection is denoted as Random, while CAM-based selection is denoted as APD. Experiments in Figure 4 show APD achieves stronger transferability than Random. This validates that leveraging CAMs to identify important regions to drop is more effective than dropping regions randomly. Using attention to guide region selection improves transferability.

**Comparison of Different Value of Hyper-parameters $\beta$.** We conduct ablation study on the hyper-parameters of the proposed method, i.e., the scale parameter $\beta$. We use Inception-v3 and ResNet101 as the surrogate models to attack other six models, respectively. And we integrate APD to MI-FGSM Dong et al. (2018) as the testing method. We test the effect of parameter $\beta$. The values of $\beta$ are set to be 3, 6, 9, 12, 15, 18, 21, 24, 27, 30, and 33. In Figure 5, we plot the attack success rate with different $\beta$s. Results show that the maximum value is obtained at $\beta = 27$ almost in all conditions, expect for when we use Inc-V3 as source model to attack Inc-v3$_{ens3}$ and Inc-v3$_{ens4}$, where the attack success rate is 0.4% and 0.5% lower than $\beta = 24$ and $\beta = 30$, respectively.

**Comparison of Different Number of Centers and Scales.** We further conduct ablation studies on the number of centers and scales, which represent the diversity of attention regions captured. Controlled experiments are performed fixing one variable while changing the other. As shown in

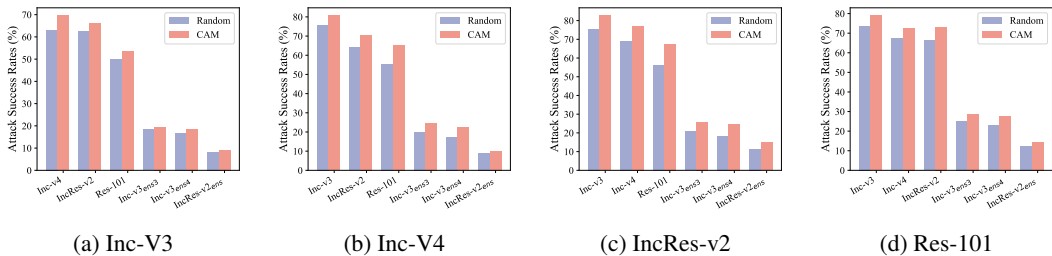

(a) Inc-V3      (b) Inc-V4      (c) IncRes-v2      (d) Res-101

Figure 4: Attack Success Rates (%) on the other six models using adversarial examples generated by Random, and APD attacks on four source models.

Figure 6, increasing the center number improves performance until stabilizing at 4, since most images contain fewer than 4 attention blocks. Similarly, increasing the scale number enhances success rate until saturating at 7, as this covers a sufficiently wide range of attention block sizes. Overall, using adequate centers and scales is key for capturing diverse attention regions, while excessive numbers provide diminishing returns.

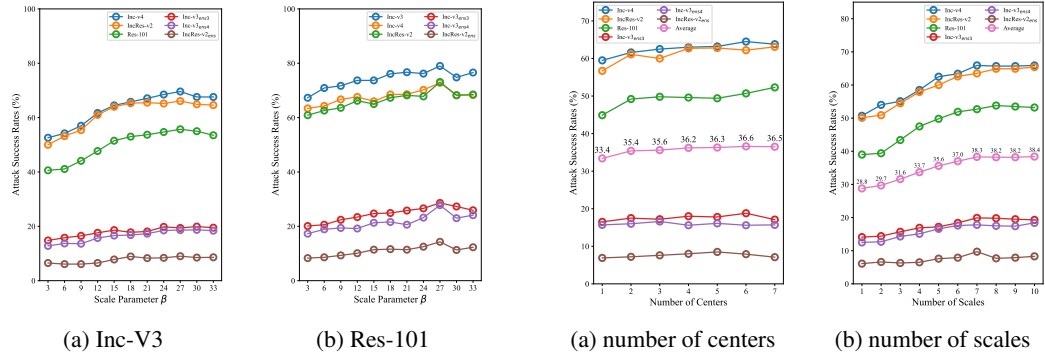

(a) Inc-V3      (b) Res-101          (a) number of centers      (b) number of scales

Figure 5: Attack success rates (%) on the other six models with adversarial examples crafted by APD attack on Inc-v3 and ResNet-101 for various number of scale factor, $\beta$.

Figure 6: Attack success rates (%) on the other six models with adversarial examples crafted by APD attack on Inc-v3 and ResNet-101 for various number of centers and scales.

**Additional Discussion and Experiments.** Since our method has additional computational cost compared to the original I-FGSM, to demonstrate that the improved transferability originates from our APD approach rather than the increased computation, we include additional discussion and experiments in the A.3 of *Appendix*.

## 5 CONCLUSION

In this study, we introduce a novel adversarial attack method, namely, the Adversarial Perturbation Dropout (APD), that can achieve significant transferability of adversarial examples. The APD method adopts the dropout mechanism on a set of adversarial images to break the synergy of the perturbations across different attention regions, which can maintain the attack effect for the target model even part of the perturbations are not in its attention regions. To improve the effectiveness of the APD attack method, we incorporate class attention maps to determine the midpoint of dropped regions with different dropped region sizes. Our approach offers a new perspective on improving transferability by reducing the interaction between different regions, which can produce robust perturbations to the target model. Through extensive experimentation, we demonstrate the superior performance of APD compared with the state-of-the-art methods. Our method can also be seamlessly integrated into existing iteration-based attack methods, which can provide great inspiration for improving the adversarial transferability. In the A.2 of *Appendix*, we also discuss why APD is effective by comparing it to the cutout attack method.

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

# A APPENDIX

## A.1 ANALYSIS OF SYNERGIES BETWEEN PERTURBATIONS

As stated above, we propose that the synergies between perturbations in those attention blocks have greater impact on transferability across models. Here, we do experiments to verify our thought. First, we use Inception-v3 (Inc-V3) Szegedy et al. (2016) as the source model to generate adversarial examples on a subset of ImageNet Russakovsky et al. (2015) containing 1000 images. Next, we apply two different perturbation removal methods to the adversarial examples. One method randomly selects a square region and removes the perturbations within that region. The other method utilizes CAM heatmaps, identifies a hotspot (local maxima) as the center, and removes perturbations in the surrounding square region. To provide stronger validation, we set square regions of various sizes. We then attack Inception-v4 (Inc-v4) Szegedy et al. (2017), Inception-Resnet-v2 (IncRes-v2) Szegedy et al. (2017) and Resnet-v2-101 (Res-101) He et al. (2016) using the adversarial examples with partially removed perturbations. As shown in Figure 7, dropping perturbations at those attention blocks can cause greater performance diminish so that we can know perturbations at those attention blocks play a greater attacking synergy with perturbations at other part.

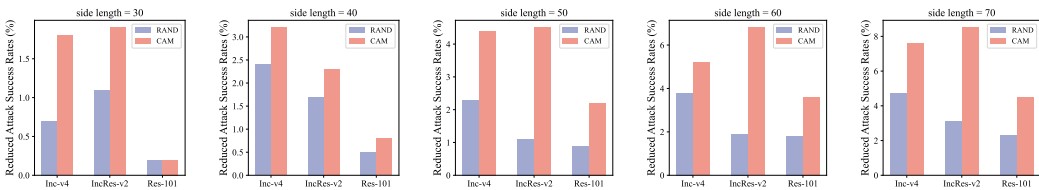

Figure 7: Reduced Attack Success Rates (%) on three tatget models.

## A.2 COMPARISON OF CUTOUT AND DROPOUT

In addition, we discuss why APD attack can enhance the transferability of adversarial attack effectively. Dropping perturbations will expose original semantic information of the object in the image, so the source model views this region as a normal, common part to recognize. So, the source model is prone to generate stronger perturbations to other parts while dropping perturbations during the adversarial example generating process. To verify this, we test the effectiveness of cutout method(i.e., using zeros to fill dropped regions) which will destroy semantic information of dropped region. As shown in Figure 8, APD showed stronger transferability than cutout way, which verifies our thinking.

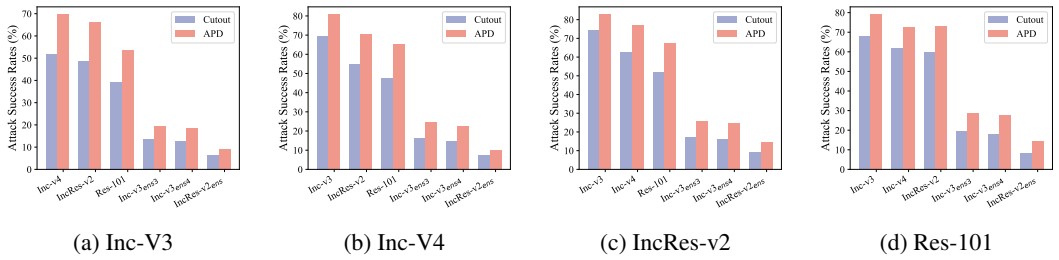

| (a) Inc-V3 | (b) Inc-V4 | (c) IncRes-v2 | (d) Res-101 |

Figure 8: Attack Success Rates (%) on the other six models using adversarial examples generated by Cutout, and APD attacks on four source models.

## A.3 COMPARISON OF SAME COMPUTATIONAL COST

Applying MI-FGSM with 1× and 15× iterations as the baseline, we display the results in Table 4, which show that increasing computational costs only slightly improve the transferability for limited cases, while the dropout method is significantly effective.

Table 4: ASR (%) results. * indicates white-box setting.

| Source Model | Method | Inc-v3 | Inc-v4 | IncRes-v2 | Res-101 | Inc-v3ens3 | Inc-v3ens4 | IncRes-v2ens |
|---|---|---|---|---|---|---|---|---|
| | MI(1x) | **100.0*** | 50.0 | 47.8 | 38.1 | 14.4 | 12.7 | 6.2 |
| Inc-v3 | MI(15x) | **100.0*** | 51.2 | 51.2 | 40.6 | 12.9 | 12.5 | 5.3 |
| | APD-MI | **100.0*** | **67.2** | **67.4** | **57.5** | **22.7** | **21.7** | **9.5** |
| | MI(1x) | 67.2 | **100.0*** | 50.5 | 45.4 | 14.3 | 12.9 | 6.5 |
| Inc-v4 | MI(15x) | 59.8 | **100.0*** | 46.8 | 41.5 | 13.5 | 11.6 | 6.2 |
| | APD-MI | **80.7** | **100.0*** | **70.7** | **65.2** | **24.6** | **22.3** | **10.2** |
| | MI(1x) | 66.2 | 57.3 | 99.9* | 46.1 | 14.2 | 14.8 | 7.9 |
| IncRes-v2 | MI(15x) | 64.7 | 53.5 | **100.0*** | 43.2 | 13.3 | 12.3 | 6.2 |
| | APD-MI | **82.7** | **77.1** | 99.8* | **67.5** | **25.9** | **24.7** | **14.8** |
| | MI(1x) | 67.6 | 61.8 | 59.8 | **99.9*** | 18.6 | 15.8 | 8.5 |
| Res-101 | MI(15x) | 69.0 | 62.0 | 60.3 | 99.7* | 16.4 | 15.4 | 7.5 |
| | APD-MI | **79.0** | **72.8** | **73.1** | 99.6* | **28.6** | **27.8** | **14.3** |

## A.4 FULL PSEUDOCODE DESCRIPTION OF THE ALGORITHM

In this section, We provide a detailed pseudocode description of our APD algorithm.

---

**Algorithm 1:** ADP Attack Algorithm

---

**Input:** A clean example $\boldsymbol{x}^{clean}$, the corresponding ground truth label $y^{true}$, a classifier $f$ with loss function $J$;

**Input:** perturbation size $\epsilon$, maximum iterations $T$, step size $\alpha$; image number for each feature region $m$;

**Input:** the function $GetCAM(f; \boldsymbol{x})$ which takes a classifier $f$ and an image $\boldsymbol{x}$ as input and outputs the class activation map on $\boldsymbol{x}$ of $f$;

**Output:** The adversarial example $\boldsymbol{x}^{adv}$.

1   $\boldsymbol{x}_0^{adv} = \boldsymbol{x}^{clean}$;          ▷ $\boldsymbol{x}^{clean}$ is $3 \times W \times H$ size tensor;

2   **for** $t = 0$ *to* $T$ **do**

3      $\boldsymbol{g} = 0$;

4      $M = GetCAM(f; \boldsymbol{x}_t^{adv})$;      ▷ $M$ is class activation map and has the size of $W \times H$;

5      Get the coordinates set of $M$'s local maximum points $Centers$, let $n = |Centers|$. $n$ is limited to less than 3;

6      **for** $j = 1$ *to* $n$ **do**

7          $r_x, r_y = Centers[j]$;

8          **for** $k = 1$ *to* $m$ **do**

9              $x_1 = max(r_x - \beta k, 0)$;

10             $x_2 = min(r_x + \beta k, W)$;

11             $y_1 = max(r_y - \beta k, 0)$;

12             $y_2 = min(r_y + \beta k, H)$;

13             $\boldsymbol{x}_{jk}^{drop} = \boldsymbol{x}_t^{adv}$;

14             $\boldsymbol{x}_{jk}^{drop}[x_1: x_2, y_1: y_2] = \boldsymbol{x}^{clean}[x_1: x_2, y_1: y_2]$;

15             Calculate the gradient $\nabla_{\boldsymbol{x}_{jk}^{drop}} J\left(\boldsymbol{x}_{jk}^{drop}, y^{true}\right)$;

16             Sum the gradients as: $\boldsymbol{g} = \boldsymbol{g} + \nabla_{\boldsymbol{x}_{jk}^{drop}} J\left(\boldsymbol{x}_{jk}^{drop}, y^{true}\right)$;

17      Get average gradients as $\boldsymbol{g} = \frac{1}{n \cdot m} \cdot \boldsymbol{g}$;

18      Update $\boldsymbol{x}_{t+1}^{adv}$ by **??**;

19   **return** $\boldsymbol{x}^{adv} = \boldsymbol{x}_T^{adv}$;

---

