# OpenReview forum: "APD: Boosting Adversarial Transferability via Perturbation Dropout"
_ICLR.cc/2024/Conference — Submitted to ICLR 2024_

### Official Review · Reviewer_3YWF · 2023-10-28

**Soundness:** 2 fair
**Presentation:** 2 fair
**Contribution:** 1 poor
**Rating:** 5
**Confidence:** 4

**Summary:**

This paper introduces a novel adversarial attack method called Adversarial Perturbation Dropout (APD), which enhances the transferability of adversarial examples. APD disrupts perturbations across different attention regions by applying dropout mechanisms to adversarial images, ensuring the attack effect for the target model even when perturbations are outside its attention regions. To enhance APD's effectiveness, class attention maps are incorporated to refine dropped regions. Extensive experiments demonstrate the effectiveness of APD.

**Strengths:**

This work's motivation is intuitive, making it overall easy to follow. The topic of this paper is valuable.

**Weaknesses:**

1. I don't think attention-based attack methods are innovative; in fact, this is a classic approach in the study of adversarial transferability. Unfortunately, the authors did not discuss its relevance to existing attention-based attack methods.
2. Why not consider the more challenging topic of targeted attacks instead of limiting the scope to non-targeted attack scenarios?
3. Why doesn't DBA compare or integrate with existing higher-performing methods[3][4]?

If the authors can provide reasonable explanations for these issues, I am inclined to increase the score.

[1] Wu W, Su Y, Chen X, et al. Boosting the transferability of adversarial samples via attention[C]//Proceedings of the IEEE/CVF Conference on Computer Vision and Pattern Recognition. 2020: 1161-1170.
[2] Wang J, Liu A, Yin Z, et al. Dual attention suppression attack: Generate adversarial camouflage in physical world[C]//Proceedings of the IEEE/CVF Conference on Computer Vision and Pattern Recognition. 2021: 8565-8574.
[3] Zhu Y, Chen Y, Li X, et al. Toward understanding and boosting adversarial transferability from a distribution perspective[J]. IEEE Transactions on Image Processing, 2022, 31: 6487-6501.
[4] Qin Z, Fan Y, Liu Y, et al. Boosting the transferability of adversarial attacks with reverse adversarial perturbation[J]. Advances in Neural Information Processing Systems, 2022, 35: 29845-29858.

**Questions:**

See above.

---

> ### Author Response · Authors · 2023-11-15
>
> ## 1. Relevance to existing attention-based attack methods
>
> Indeed, you're right. Attention-based attack methods have been a classic tool within the realm of studying adversarial transferability. However, it's essential to note that the attention-based method is a commonly used technique in computer vision, especially in semantic segmentation, serving as a fundamental method. First and foremost, it needs to be clarified that existing methods(including ours) do not use attention as innovative point itself but rather utilize it as a tool. So, we did not specifically discuss attention in our paper, but rather focused on noise effectiveness of I-FGSM, TI-FGSM, and PI-FGSM.
>
> Below, I will introduce the relationship between our method and existing methods[1][2][3] using attention.
>
> [1] Evading defenses to transferable adversarial examples by translation-invariant attacks. CVPR 2019.
> [2] Boosting the transferability of adversarial samples via attention. CVPR 2020.
> [3] Dual attention suppression attack: Generate adversarial camouflage in physical world. CVPR 2021.
>
> ### 1.1 Evading defenses to transferable adversarial examples by translation-invariant attacks.
>
> In our Section 1, i.e., Introduction, we have already discussed it.
>
> ### 1.2 Boosting the transferability of adversarial samples via attention.
>
> "Boosting the transferability of adversarial samples via attention" is highly relevant to our approach, we were not aware of it previously. We appreciate your sharing this paper and will include relevant discussions in the final paper.
>
> **1) Their method**, illustrated in Figure 1 (refer to their paper), highlights attention heatmaps of various models on a cat image, primarily focusing on the cat's face. They observe that most models extract similar important features, leading to their method's design to **increase the perturbation norm in these important features' regions while reducing the perturbation norm in unimportant regions**.
>
> **2) Our method:** We observe that not all models focus on the same position, indicating their observation is somewhat insufficient. Our Figure 2(refer to our paper) displays a turtle image, which could have multiple important features, and different models might identify different ones. **Our method decouples different regions, enhancing their independence** to improve transferability.
>
> **3) Comparision:** Their method succeeds because they reduce the norm of noise in unimportant regions while increasing the norm of noise in important regions. In contrast, our method succeeds because it decouples the effects of noise on different blocks in important regions. In principle, the two methods are inconsistent.
>
> **4) More Discussion:** Inspired by this paper, we also discover a weakness of our method, but we demonstrate that this weakness has an almost negligible impact on our method. If an image aligns with their cat image, which just has one important region and all models share attention on it, looks like our method's gonna lose its powers. Specifically, our method can only decouple the background from the important feature region, which will not improve transferability significantly in this situation. To explain this, we utilize different models to generate CAM on the dataset and compute the Number of Hotspot Regions for each image. Next, we count the Number of Images that own specific Number of Hotspot Regions, as shown in **Table [1]**.

---

> > ### Author Response · Authors · 2023-11-15
> >
> > ### Table[1]: Distribution of Images by Number of Hotspot Regions
> >
> > #### Inc-v3
> >
> > | Number of Hotspot Regions | 0  | 1  | 2   | 3   | 4   | 5   | 6   | 7   | 8   | 9   | 10  | 11  | 12  | 13  | 16  |
> > |---------------------------|----|----|-----|-----|-----|-----|-----|-----|-----|-----|-----|-----|-----|-----|-----|
> > | Number of Images          | 1  | 30 | 99  | 157 | 176 | 170 | 141 | 91  | 58  | 31  | 28  | 7   | 5   | 5   | 1   |
> >
> > #### Inc-v4
> >
> > | Number of Hotspot Regions | 0  | 1  | 2   | 3   | 4   | 5   | 6   | 7   | 8   | 9   | 10  | 11  | 12  | 14  | 15  |
> > |---------------------------|----|----|-----|-----|-----|-----|-----|-----|-----|-----|-----|-----|-----|-----|-----|
> > | Number of Images          | 1  | 31 | 100 | 178 | 194 | 175 | 139 | 79  | 51  | 23  | 15  | 10  | 2   | 1   | 1   |
> >
> > #### Inc-Res-v2
> >
> > | Number of Hotspot Regions | 0  | 1  | 2   | 3   | 4   | 5   | 6   | 7   | 8   | 9   | 10  | 11  | 12  | 13  | 14  | 15  | 16  | 17  |
> > |---------------------------|----|----|-----|-----|-----|-----|-----|-----|-----|-----|-----|-----|-----|-----|-----|-----|-----|-----|
> > | Number of Images          | 1  | 18 | 51  | 126 | 169 | 179 | 191 | 103 | 56  | 46  | 21  | 16  | 7   | 6   | 5   | 3   | 1   | 1   |
> >
> > #### Res-101
> >
> > | Number of Hotspot Regions | 0   | 2   | 3   | 4   | 5   | 6   | 7   | 8   | 9   | 10  | 11  | 12  | 13  | 14  | 15  | 16  | 17  | 18  | 19  | 20  | 21  | 22  | 23  | 24  |
> > |---------------------------|-----|-----|-----|-----|-----|-----|-----|-----|-----|-----|-----|-----|-----|-----|-----|-----|-----|-----|-----|-----|-----|-----|-----|-----|
> > | Number of Images          | 19  | 2   | 13  | 17  | 34  | 45  | 47  | 102 | 90  | 91  | 107 | 109 | 94  | 79  | 61  | 33  | 15  | 20  | 8   | 6   | 4   | 1   | 1   | 2   |
> >
> > ### 1.3 Dual attention suppression attack: Generate adversarial camouflage in the physical world.
> >
> > "Dual attention suppression attack: Generate adversarial camouflage in the physical world," differs significantly from our method. The only commonality lies in the use of attention heatmaps as a tool. **The Dual Attention Suppression (DAS) attack generates visible physical-world adversarial camouflage by redirecting model attention from the target to non-target regions to achieve misclassification. This is done by gudience of attention.** It focuses on generating physical adversarial examples, and both the principle and the target task differ substantially from our work.

---

> ### Author Response · Authors · 2023-11-15
>
> ## 2. Targeted attacks
>
> We chose not to conduct targeted attack because we consider it a more specialized task that requires dedicated methods to solve. For instance, as indicated in the appendix of the RAP [1] method, on page 22, there's a table for Inc-v3 → ResNet-50 target attack. The best attack achieved only a 16.7% attack success rate, and some fundamental methods had a mere 0.1% attack success rate. In our experiment, we used models that were more complex and difficult to attack than theirs, including Inc-v3, Inc-v4, IncRes-v2 and Res-101, the targeted attack success rates among these models do not exceed 0.1% using MI-FGSM.
>
> However, we do test our method's effectiveness using an method specifically designed for target attacks, namely CFM[2]. Combining our method with this specialized attack for target attacks can improve attack success rate significantly. Here are our results in **Table[2]**. CFM-RDI-MI is their SOTA attack in their paper and APD-CFM-RDI-MI is our attack, which achieves new SOTA results.
>
> [1] Boosting the transferability of adversarial attacks with reverse adversarial perturbation. NeurIPS 2022.
> [2] Introducing Competition to Boost the Transferability of Targeted Adversarial Examples through Clean Feature Mixup. CVPR 2023.
>
> ### Table[2]: Attack Success Rate(\%) of target attack
>
> | Source Model | Attack Method         | Inc-v3 | Inc-v4 | IncRes-v2 | Res-101 |
> |--------------|-----------------------|--------|--------|-----------|---------|
> | Inc-v3       | CFM-RDI-MI         | 98.3   | 43.4   | 34.4      | 27.5    |
> | Inc-v3             | APD-CFM-RDI-MI     | 98.3   | 68.5   | 59.8      | 53.6    |
> | Inc-v4       | CFM-RDI-MI         | 30.7   | 98.6   | 30.3      | 15.9    |
> | Inc-v4             | APD-CFM-RDI-MI     | 55.6   | 98.9   | 53.4      | 36.5    |
> | IncRes-v2    | CFM-RDI-MI         | 14.3   | 23.4   | 99.7      | 8.4     |
> | IncRes-v2             | APD-CFM-RDI-MI     | 34.0   | 42.5   | 99.8      | 19.2    |
> | Res-101      | CFM-RDI-MI         | 50.7   | 50.3   | 29.4      | 98.2    |
> | Res-101             | APD-CFM-RDI-MI     | 56.1   | 55.6   | 35.4      | 98.2    |

---

> ### Author Response · Authors · 2023-11-15
>
> ## 3. Comparisons with more relevant methods
>
> Thank you for your valuable suggestion. We have incorporated additional comparisons with advanced methods, namely PI-FGSM[1], RAP[2], SSA[3], PGN[4], TAIG[5], and CFM[6]. Partial results are presented in **Table[3]**, with the remaining outcomes slated for inclusion in the final version of our paper. Note that we have not included the method "Toward understanding and boosting adversarial transferability from a distribution perspective," as provided by you. This is due to the significant time consumption associated with their approach, as indicated in their paper: "The computation cost of fine-tuning the surrogate model for one epoch requires 8 hours on Tesla V100 using ResNet-50 on ImageNet." Therefore, we require additional time and plan to incorporate comparative results with this method in the final version of the paper.
>
> We appreciate your feedback and believe these enhancements strengthen the comprehensiveness of our comparative analysis.
>
> [1] PI-FGSM: Patch-wise Attack for Fooling Deep Neural Network. ECCV 2020.
>
> [2] Boosting the transferability of adversarial attacks with reverse adversarial perturbation. NeurIPS 2022.
>
> [3] Frequency Domain Model Augmentation for Adversarial Attack. ECCV 2022.
>
> [4] Boosting Adversarial Transferability by Achieving Flat Local Maxima. NeurIPS 2023.
>
> [5] Transferable Adversarial Attack based on Integrated Gradients. ICLR 2022.
>
> [6] Introducing Competition to Boost the Transferability of Targeted Adversarial Examples through Clean Feature Mixup. CVPR 2023.
>
> ### Table[3]: Attack Success Rates (\%) of adversarial attacks. We integrate our method into all the baselines, such as we let 'PI/+APD' denote PI and APD-PI. The best results are bold. * indicates the white-box setting.
>
> Note that for all methods, we integrate momentum because we take momentum as a fundamental method.
>
> #### Crafted on Inc-v3
>
> | Attack   | Inc-v3*          | Inc-v4           | IncRes-v2        | Res-101          | Inc-v3$_{ens3}$  | Inc-v3$_{ens4}$  | IncRes-v2$_{ens}$ |
> |----------|------------------|------------------|------------------|------------------|------------------|------------------|------------------|
> | PI/+APD  | **100.0/100.0**  | 56.3/**63.2**    | 51.6/**58.6**    | 45.2/**51.1**    | 21.5/**24.0**    | 20.1/**23.3**    | 11.5/**13.3**    |
> | RAP/+APD | **100.0/100.0**  | 50.1/**63.8**    | 48.8/**63.9**    | 39.2/**52.8**    | 15.1/**17.7**    | 13.7/**18.5**    | 6.4/**9.0**      |
> | SSA/+APD | **99.6/99.6**    | 54.9/**66.3**    | 53.7/**65.2**    | 48.9/**59.7**    | 15.3/**21.4**    | 15.7/**21.2**    | 6.7/**10.8**     |
> | TAIG-S/+APD | **100.0/100.0** | 75.6/**86.1**    | 75.0/**85.5**    | 66.6/**80.0**    | 22.7/**34.1**    | 21.9/**32.9**    | 12.6/**17.4**    |
> | PGN/+APD | 98.7/**98.9**    | 63.3/**73.4**    | 59.2/**69.6**    | 53.4/**63.8**    | 20.4/**27.4**    | 20.7/**25.9**    | 8.9/**11.9**     |
> | CFM/+APD | **100.0/100.0**  | 34.6/**49.7**    | 28.1/**44.3**    | 69.7/**78.9**    | 17.0/**21.0**    | 15.0/**19.7**    | 10.7/**13.2**    |
>
>
> #### Crafted on Inc-v4
>
> | Attack   | Inc-v3 | Inc-v4* | IncRes-v2 | Res-101 | Inc-v3$_{ens3}$ | Inc-v3$_{ens4}$ | IncRes-v2$_{ens}$ |
> |----------|--------|---------|-----------|---------|------------------|------------------|-------------------|
> | PI/+APD  | 68.3/**73.5** | **100.0/100.0** | 55.2/**61.3** | 47.6/**54.8** | 23.3/**26.5** | 22.3/**25.5** | 15.2/**16.2** |
> | RAP/+APD | 65.1/**79.9** | **100.0/100.0** | 51.1/**67.3** | 45.8/**60.6** | 15.1/**21.9** | 14.2/**18.6** | 6.6/**10.2** |
> | SSA/+APD | 73.9/**80.9** | 98.6/**99.0** | 60.2/**70.8** | 58.4/**68.9** | 19.1/**28.5** | 17.6/**25.5** | 8.9/**14.4** |
> | TAIG-S/+APD | 80.9/**91.7** | **100.0**/99.9 | 72.4/**87.2** | 67.4/**81.2** | 23.5/**37.9** | 21.7/**34.0** | 13.1/**21.2** |
> | PGN/+APD | 81.6/**87.0** | **99.6** /98.9| 68.5/**80.6** | 64.6/**74.3** | 25.8/**35.1** | 25.3/**32.1** | 13.3/**17.4** |
> | CFM/+APD | 76.9/**88.1** | **100.0/100.0** | 32.6/**56.1** | 72.4/**82.8** | 17.9/**28.0** | 16.9/**25.5** | 11.7/**17.7** |

---

> ### Author Response · Authors · 2023-11-15
>
> #### Crafted on IncRes-v2
>
> | Attack   | Inc-v3          | Inc-v4          | IncRes-v2*      | Res-101         | Inc-v3$_{ens3}$ | Inc-v3$_{ens4}$ | IncRes-v2$_{ens}$ |
> |----------|-----------------|-----------------|-----------------|-----------------|-----------------|-----------------|-------------------|
> | PI/+APD  | 69.9/**78.7**   | 65.6/**72.6**   | **100.0/100.0** | 51.8/**63.2**   | 20.8/**25.3**   | 17.7/**23.7**   | 12.1/**15.1**    |
> | RAP/+APD | 66.4/**80.1**   | 56.9/**72.9**   | **100.0**/99.8  | 45.3/**62.3**   | 15.6/**24.6**   | 13.2/**21.2**   | 7.0/**13.7**     |
> | SSA/+APD | 82.7/**88.0**   | 71.3/**80.2**   | 98.9/**99.1**   | 67.1/**74.6**   | 23.1/**34.5**   | 20.4/**29.4**   | 10.8/**17.2**    |
> | TAIG-S/+APD | 89.0/**94.6**   | 85.5/**91.7**   | **100.0/100.0** | 78.5/**88.0**   | 28.5/**48.2**   | 25.8/**43.1**   | 17.3/**32.3**    |
> | PGN/+APD | 87.2/**90.7**   | 80.0/**83.5**   | **99.1**/98.7   | 73.1/**80.0**   | 31.6/**43.6**   | 28.3/**37.2**   | 15.7/**23.5**    |
> | CFM/+APD | 84.9/**92.0**   | 61.8/**78.1**   | **100.0/100.0** | 80.3/**86.9**   | 24.4/**39.3**   | 21.7/**34.1**   | 16.4/**27.6**    |
>
>
> #### Crafted on Res-101
>
> | Attack   | Inc-v3 | Inc-v4 | IncRes-v2 | Res-101* | Inc-v3$_{ens3}$ | Inc-v3$_{ens4}$ | IncRes-v2$_{ens}$ |
> |----------|--------|--------|-----------|----------|------------------|------------------|-------------------|
> | PI/+APD  | 70.2/**76.7** | 63.6/**68.7** | 61.5/**69.7** | **100.0/100.0** | 26.7/**32.4** | 25.3/**30.3** | 14.5/**18.4** |
> | RAP/+APD | 68.4/**78.6** | 62.8/**69.7** | 59.0/**69.4** | **99.6/99.6** | 20.6/**27.0** | 16.9/**24.0** | 8.5/**12.4** |
> | SSA/+APD | 80.0/**81.3** | 69.1/**72.8** | 68.5/**71.6** | **99.6/99.6** | 25.8/**33.8** | 22.4/**29.7** | 11.9/**15.1** |
> | TAIG-S/+APD | 79.7/**84.9** | 74.1/**81.5** | 72.8/**79.6** | **100.0/100.0** | 30.1/**39.8** | 26.3/**35.4** | 16.1/**22.9** |
> | PGN/+APD | 84.5/**84.8** | 76.1/**77.9** | 77.5/**76.4** | 99.8/**100.0** | 34.7/**41.2** | 30.7/**36.9** | 17.6/**20.8** |
> | CFM/+APD | 83.7/**87.8** | 60.2/**65.1** | 50.2/**58.4** | **100.0/100.0** | 22.4/**28.3** | 22.3/**27.8** | 14.7/**19.3** |

---

> ### Author Response · Authors · 2023-11-22
>
> Dear Reviewer 3YWF,
>
> Thank you again for your time. As the deadline for discussion is approaching, we do wish to hear from you to see if our response resolves your concerns. We are happy to provide any additional clarifications if needed.

---

### Official Review · Reviewer_e8ZY · 2023-10-29

**Soundness:** 2 fair
**Presentation:** 3 good
**Contribution:** 2 fair
**Rating:** 5
**Confidence:** 4

**Summary:**

This work introduces a dropout scheme on perturbation to improve the adversarial transferability of attacks. Instead of random dropout, the authors propose to utilize CAM to discover the key points of images, which forms the dropped regions. The evaluation is conducted on ImageNet with various networks. The comparison with baselines as well as ablation studies show the effectiveness of the proposed algorithm.

**Strengths:**

1. The paper is well-organized and easy to follow.
2. The proposed dropout scheme seems to be an interesting way to tackle adversarial transferability since vanilla dropout in DNNs helps generalization.
3. The evaluation on ImageNet shows promising results and the ablation studies are sufficient.

**Weaknesses:**

1. There seems to be a gap between CAM regions and the dropped regions in Figure 3. However, the reason why the proposed algorithm utilizes midpoints with square regions instead of the CAM regions is not well-explained.
2. The complexity needs more discussion since the proposed algorithm requires more iterations, such as the time consumption comparison with other baselines in Table 1.
3. More evaluation on other advanced vision models, such as vision transformer [a, b].
4. More comparison with other adversarial transferability works on attention or regularization [c, d].

Minors:
1. Missing definition of $\alpha$ in Section 3.2.
2. Undefined Eq in Line 18 of Algorithm 1.

[a]. Transferable Adversarial Attack for Both Vision Transformers and Convolutional Networks via Momentum Integrated Gradients. ICCV 2023.

[b]. Transferable Adversarial Attacks on Vision Transformers with Token Gradient Regularization. CVPR 2023.

[c]. Improving Adversarial Transferability via Neuron Attribution-based Attacks. CVPR 2022.

[d]. Boosting the transferability of adversarial attacks with reverse adversarial perturbation. NeurIPS 2022.

**Questions:**

1.	Please clarify the gap in Figure 3.
2.	Please discuss the complexity.
3.	Please provide more evaluation on advanced vision models.
4.	Please include more comparisons.

---

> ### Author Response · Authors · 2023-11-15
>
> ## 1. Why do we use hotspots instead of using thresholds to determine the dropped regions.
>
> Thank you for bringing this to our attention. We appreciate your feedback, and we acknowledge that we inadvertently overlooked an important aspect. We will include the following clarification in our paper:
>
> In our motivation experiment depicted in Fig1(b), we employed a threshold to identify regions where the source model focuses while the target model does not. However, in our formal method, we opted to utilize midpoints with square regions.
>
> Here's the rationale behind this choice:
>
> 1. **Search for Effective Dropout Method**: Our goal is to discover a more effective dropout method by exploring regions where the synergy is more pronounced across models.
> 2. **Observation from Experiment (Fig1(b))**: Through our experiments, especially in Fig1(b), we found that the "regions where the source model focuses while the target model does not" have a significant negative impact on transferability. This observation led us to conclude that differences in attention regions between the two models contribute most to the negative impact on transferability.
>
> 3. **Identification of Attention Region Differences**: Combining the findings from points 1 and 2, it became crucial to identify the differences in attention regions across different models. From our observations in Fig2, we noted that a model's attention region comprises some meaningful blocks. By decoupling perturbations in these different blocks, we can effectively improve transferability.
>
> 4. **Challenges with Threshold-Based Approach**: When using a threshold to choose dropped regions, two challenges arise: i) if the threshold is set too low, different blocks merge together, and ii) if set too high, each dropped region does not cover one entire block. Additionally, different attention regions have different values, making it challenging to determine an optimal threshold. In contrast, the hotspot method accurately identifies each block. Therefore, we opted for the hotspot midpoint approach. The use of a square region is more of a customary choice. Shapes like circles or triangles are also workable. The emphasis is on choosing the hotspot's center rather than fixating on the shape.
>
> ## 2. Time consumption comparison.
>
> Here is our analysis of Computational Cost:
>
> In our proposed method, the time cost primarily arises from forward propagation, backward propagation, and input transformation. Notably, the computation process of the CAM heatmap can be seamlessly integrated into the forward propagation, rendering the heatmap computation time negligible. As the primary time cost is associated with forward and backward calculations, we omit other factors like input transformation.
>
> Algorithm 1(see in our paper) involves a global iteration number T, implying 2T forward and backward calculations, local maximum points number n, and size scale number m. Utilizing the parameters from MI, DIM, TIM, SIM, and AAM, we summarize the time cost relative to MI, where MI's cost is considered as 1x:
>
> | Method   | Time Cost |
> |----------|-----------|
> | MI       | 1x        |
> | DIM      | 1x        |
> | TIM      | 1x        |
> | SIM      | 5x        |
> | AAM      | 15x       |
> | APDM(ours) | <15x (since $n$ is less than or equal to 3 in our experiments) |
>
> The process of generating adversarial examples is conducted offline. As the primary evaluation metric is the attack success rate, with time cost being a secondary consideration, we have provided additional discussion and experiments in Appendix A.3 of our original paper. Given that our method incurs additional computational cost compared to the original MI-FGSM, we aim to illustrate in the A.3 section that the improved transferability results from our dropout mechanism rather than the increased computational cost.
>
> We appreciate your consideration and are open to any further questions or clarifications.

---

> ### Author Response · Authors · 2023-11-15
>
> ## 3. Evaluation on diversity of model architectures
>
> Thank you for highlighting this concern. We appreciate your suggestion and would like to inform you that our paper already includes an evaluation of the diversity of model architectures. In Table 3 of our original paper, we considered various network architectures, including sequencer deep LSTM, ViT-B/16, and MnasNet.
>
> Upon reviewing the paper you provided, we recognized the opportunity to further enhance the diversity of our model evaluations. As a result, we have expanded our architecture diversity by incorporating additional models, including Inc-v3, Inc-v4, IncRes-v2, Res-101, ViT-B/16, TNT, Swin, PiT-B, CaiT-S/24, Visformer-S, and SNNs (Spiking Neural Networks). It's worth noting that we selected four CNN-based models (Inc-v3, Inc-v4, IncRes-v2, Res-101), six Transformer-based models (ViT-B/16, TNT, Swin, PiT-B, CaiT-S/24, Visformer-S), and introduced SNNs to broaden the spectrum of architecture diversity. Deep learning, as a broader category, encompasses artificial neural networks (ANNs). SNNs, on the other hand, represent a distinct type of neural network within the realm of artificial neural networks.
>
> To provide you with a glimpse of the results, we have included part of the findings in Table[1]. Rest assured, we will incorporate the complete set of results into our paper in the final paper:
>
> ### Table[1] Attack Success Rate(\%) of models of diverse architectures
> | Source Model | Attack Method | Inc-v3 | Inc-v4 | IncRes-v2 | Res-101 | ViT-B/16 | TNT | Swin | PiT-B | CaiT-S/24 | Visformer-S | SNNs |
> |--------------|---------------|--------|--------|----------|---------|----------|-----|------|-------|-----------|-------------|------|
> | Inc-v3       | MI            | 100.00 | 49.20  | 47.10    | 36.90  | 30.10    | 32.20 | 12.30 | 14.10 | 12.40     | 19.10       | 55.00 |
> | Inc-v3       | APD-MI        | 100.00 | 68.10  | 65.70    | 56.30  | 36.10    | 41.60 | 19.30 | 27.60 | 21.70     | 30.30       | 69.70 |
> | IncRes-v2    | MI            | 66.40  | 57.40  | 99.80    | 44.10  | 33.60    | 38.00 | 12.50 | 17.10 | 13.80     | 22.90       | 60.20 |
> | IncRes-v2     | APD-MI        | 81.60  | 76.00  | 99.90    | 64.70  | 40.70    | 50.40 | 25.00 | 35.50 | 29.10     | 38.00       | 75.00 |
> | ViT-B/16     | MI            | 37.80  | 30.40  | 28.60    | 33.80  | 100.00   | 46.50 | 22.90 | 23.80 | 22.70     | 27.20       | 50.90 |
> | ViT-B/16             | APD-MI        | 44.23  | 40.16  | 34.28    | 41.52  | 100.00   | 58.26 | 39.82 | 39.37 | 35.63     | 40.72       | 56.90 |
> | Swin         | MI            | 43.90  | 36.70  | 33.60    | 36.10  | 41.70    | 47.70 | 99.80 | 35.40 | 30.60     | 40.50       | 52.70 |
> |  Swin            | APD-MI        | 44.90  | 42.90  | 35.60    | 38.20  | 49.50    | 53.10 | 99.90 | 54.30 | 44.60     | 54.40       | 53.90 |
> | CaiT-S/24    | MI            | 64.40  | 57.90  | 53.30    | 55.40  | 57.30    | 89.90 | 45.50 | 61.10 | 100.00    | 68.40       | 80.80 |
> | CaiT-S/24             | APD-MI        | 67.50  | 61.20  | 59.20    | 62.60  | 64.30    | 92.70 | 66.50 | 77.20 | 100.00    | 82.30       | 83.30 |
> | SNNs         | MI            | 52.50  | 49.50  | 39.90    | 41.00  | 39.10    | 41.30 | 14.20 | 20.60 | 19.90     | 29.70       | 100.00 |
> |  SNNs            | APD-MI        | 71.30  | 67.30  | 58.70    | 60.60  | 47.80    | 59.10 | 24.60 | 38.10 | 35.50     | 49.40       | 100.00 |

---

> ### Author Response · Authors · 2023-11-15
>
> ## 4. Comparisons with more relevant methods
>
> Thank you for your valuable suggestion. We have incorporated additional comparisons with advanced methods, namely PI-FGSM[1], RAP[2], SSA[3], PGN[4], TAIG[5], and CFM[6]. Partial results are presented in Table[2], with the remaining outcomes slated for inclusion in the final version of our paper. It's worth noting that we opted not to include "Improving Adversarial Transferability via Neuron Attribution-based Attacks," as this work pertains to feature-level attacks. Furthermore, we have already integrated a more advanced feature-level attack[6].
>
> We appreciate your feedback and believe these enhancements strengthen the comprehensiveness of our comparative analysis.
>
> [1] PI-FGSM: Patch-wise Attack for Fooling Deep Neural Network. ECCV 2020.
>
> [2] Boosting the transferability of adversarial attacks with reverse adversarial perturbation. NeurIPS 2022.
>
> [3] Frequency Domain Model Augmentation for Adversarial Attack. ECCV 2022.
>
> [4] Boosting Adversarial Transferability by Achieving Flat Local Maxima. NeurIPS 2023.
>
> [5] Transferable Adversarial Attack based on Integrated Gradients. ICLR 2022.
>
> [6] Introducing Competition to Boost the Transferability of Targeted Adversarial Examples through Clean Feature Mixup. CVPR 2023.
>
> ### Table[2]: Attack Success Rates (\%) of adversarial attacks. We integrate our method into all the baselines, such as we let 'PI/+APD' denote PI and APD-PI. The best results are bold. * indicates the white-box setting.
>
> Note that for all methods, we integrate momentum because we take momentum as a fundamental method.
>
> #### Crafted on Inc-v3
>
> | Attack   | Inc-v3*          | Inc-v4           | IncRes-v2        | Res-101          | Inc-v3$_{ens3}$  | Inc-v3$_{ens4}$  | IncRes-v2$_{ens}$ |
> |----------|------------------|------------------|------------------|------------------|------------------|------------------|------------------|
> | PI/+APD  | **100.0/100.0**  | 56.3/**63.2**    | 51.6/**58.6**    | 45.2/**51.1**    | 21.5/**24.0**    | 20.1/**23.3**    | 11.5/**13.3**    |
> | RAP/+APD | **100.0/100.0**  | 50.1/**63.8**    | 48.8/**63.9**    | 39.2/**52.8**    | 15.1/**17.7**    | 13.7/**18.5**    | 6.4/**9.0**      |
> | SSA/+APD | **99.6/99.6**    | 54.9/**66.3**    | 53.7/**65.2**    | 48.9/**59.7**    | 15.3/**21.4**    | 15.7/**21.2**    | 6.7/**10.8**     |
> | TAIG-S/+APD | **100.0/100.0** | 75.6/**86.1**    | 75.0/**85.5**    | 66.6/**80.0**    | 22.7/**34.1**    | 21.9/**32.9**    | 12.6/**17.4**    |
> | PGN/+APD | 98.7/**98.9**    | 63.3/**73.4**    | 59.2/**69.6**    | 53.4/**63.8**    | 20.4/**27.4**    | 20.7/**25.9**    | 8.9/**11.9**     |
> | CFM/+APD | **100.0/100.0**  | 34.6/**49.7**    | 28.1/**44.3**    | 69.7/**78.9**    | 17.0/**21.0**    | 15.0/**19.7**    | 10.7/**13.2**    |
>
>
> #### Crafted on Inc-v4
>
> | Attack   | Inc-v3 | Inc-v4* | IncRes-v2 | Res-101 | Inc-v3$_{ens3}$ | Inc-v3$_{ens4}$ | IncRes-v2$_{ens}$ |
> |----------|--------|---------|-----------|---------|------------------|------------------|-------------------|
> | PI/+APD  | 68.3/**73.5** | **100.0/100.0** | 55.2/**61.3** | 47.6/**54.8** | 23.3/**26.5** | 22.3/**25.5** | 15.2/**16.2** |
> | RAP/+APD | 65.1/**79.9** | **100.0/100.0** | 51.1/**67.3** | 45.8/**60.6** | 15.1/**21.9** | 14.2/**18.6** | 6.6/**10.2** |
> | SSA/+APD | 73.9/**80.9** | 98.6/**99.0** | 60.2/**70.8** | 58.4/**68.9** | 19.1/**28.5** | 17.6/**25.5** | 8.9/**14.4** |
> | TAIG-S/+APD | 80.9/**91.7** | **100.0**/99.9 | 72.4/**87.2** | 67.4/**81.2** | 23.5/**37.9** | 21.7/**34.0** | 13.1/**21.2** |
> | PGN/+APD | 81.6/**87.0** | **99.6**/98.9 | 68.5/**80.6** | 64.6/**74.3** | 25.8/**35.1** | 25.3/**32.1** | 13.3/**17.4** |
> | CFM/+APD | 76.9/**88.1** | **100.0/100.0** | 32.6/**56.1** | 72.4/**82.8** | 17.9/**28.0** | 16.9/**25.5** | 11.7/**17.7** |
>
> #### Crafted on IncRes-v2
>
> | Attack   | Inc-v3          | Inc-v4          | IncRes-v2*      | Res-101         | Inc-v3$_{ens3}$ | Inc-v3$_{ens4}$ | IncRes-v2$_{ens}$ |
> |----------|-----------------|-----------------|-----------------|-----------------|-----------------|-----------------|-------------------|
> | PI/+APD  | 69.9/**78.7**   | 65.6/**72.6**   | **100.0/100.0** | 51.8/**63.2**   | 20.8/**25.3**   | 17.7/**23.7**   | 12.1/**15.1**    |
> | RAP/+APD | 66.4/**80.1**   | 56.9/**72.9**   | **100.0**/99.8  | 45.3/**62.3**   | 15.6/**24.6**   | 13.2/**21.2**   | 7.0/**13.7**     |
> | SSA/+APD | 82.7/**88.0**   | 71.3/**80.2**   | 98.9/**99.1**   | 67.1/**74.6**   | 23.1/**34.5**   | 20.4/**29.4**   | 10.8/**17.2**    |
> | TAIG-S/+APD | 89.0/**94.6**   | 85.5/**91.7**   | **100.0/100.0** | 78.5/**88.0**   | 28.5/**48.2**   | 25.8/**43.1**   | 17.3/**32.3**    |
> | PGN/+APD | 87.2/**90.7**   | 80.0/**83.5**   | **99.1**/98.7   | 73.1/**80.0**   | 31.6/**43.6**   | 28.3/**37.2**   | 15.7/**23.5**    |
> | CFM/+APD | 84.9/**92.0**   | 61.8/**78.1**   | **100.0/100.0** | 80.3/**86.9**   | 24.4/**39.3**   | 21.7/**34.1**   | 16.4/**27.6**    |

---

> > ### Author Response · Authors · 2023-11-15
> >
> > #### Crafted on Res-101
> >
> > | Attack   | Inc-v3 | Inc-v4 | IncRes-v2 | Res-101* | Inc-v3$_{ens3}$ | Inc-v3$_{ens4}$ | IncRes-v2$_{ens}$ |
> > |----------|--------|--------|-----------|----------|------------------|------------------|-------------------|
> > | PI/+APD  | 70.2/**76.7** | 63.6/**68.7** | 61.5/**69.7** | **100.0/100.0** | 26.7/**32.4** | 25.3/**30.3** | 14.5/**18.4** |
> > | RAP/+APD | 68.4/**78.6** | 62.8/**69.7** | 59.0/**69.4** | **99.6/99.6** | 20.6/**27.0** | 16.9/**24.0** | 8.5/**12.4** |
> > | SSA/+APD | 80.0/**81.3** | 69.1/**72.8** | 68.5/**71.6** | **99.6/99.6** | 25.8/**33.8** | 22.4/**29.7** | 11.9/**15.1** |
> > | TAIG-S/+APD | 79.7/**84.9** | 74.1/**81.5** | 72.8/**79.6** | **100.0/100.0** | 30.1/**39.8** | 26.3/**35.4** | 16.1/**22.9** |
> > | PGN/+APD | 84.5/**84.8** | 76.1/**77.9** | 77.5/**76.4** | 99.8/**100.0** | 34.7/**41.2** | 30.7/**36.9** | 17.6/**20.8** |
> > | CFM/+APD | 83.7/**87.8** | 60.2/**65.1** | 50.2/**58.4** | **100.0/100.0** | 22.4/**28.3** | 22.3/**27.8** | 14.7/**19.3** |
> >
> > ## 5. About Minors
> >
> > Thank you for your careful review. $\alpha$ is the step size. And we have changed Eq in Line 18 of Algorithm 1 in our paper. Later we will update our paper.

---

> ### Author Response · Authors · 2023-11-22
>
> Dear Reviewer e8ZY,
>
> Thank you again for your time. As the deadline for discussion is approaching, we do wish to hear from you to see if our response resolves your concerns. We are happy to provide any additional clarifications if needed.

---

### Official Review · Reviewer_PxjZ · 2023-10-30

**Soundness:** 2 fair
**Presentation:** 3 good
**Contribution:** 2 fair
**Rating:** 5
**Confidence:** 4

**Summary:**

The paper proposed a novel adversarial attack named adversarial Perturbation Dropout (APD), which can enhance the transferability of adversarial examples with dropout during optimization. It is activated by the investigation that attention regions are not consistent across different models and perturbations in the neglected regions also have a significant effect on transferability due to the synergy between perturbations from different regions. Simple experiments are conducted to verify the mutual influence between perturbations. APD breaks the synergy by leveraging the class activation map (CAM) to dropout perturbations. Experiments on ImageNet show that APD can achieve high transferability and efficiency across CNNs.

**Strengths:**

1. The paper presents a novel finding that perturbations in attention and neglected regions have a mutual influence, creating a synergy that reduces the transferability of attack methods.

2. A major strength of APD lies in its simplicity and effectiveness, leveraging CAM to dropout perturbations.

3. Apart from APD’s performance on transferability, the paper also considers performance on defense, further highlighting the effectiveness of the method.

4. The paper's ablation study is comprehensive, including a random drop of regions and the selection of crucial hyperparameters.

**Weaknesses:**

1. The technical novelty of perturbation dropout is limited, as similar thoughts have been proposed in recent works [1].

2. The attention regions may shift during the attack iterations; therefore, it is unreliable to rely on the CAM from the previous iteration as guidance for the current one.

3. The experiment compares AA-TI-DIM(SOTA) and some classic transfer-based attacks with APD. It would be better if comparisons with relevant methods such as the previously mentioned PI-attack that extends the perturbation to cover the object, or other dropout methods, can be made.

4. In order to verify the mutual influence between perturbations from different regions, the paper compares random noise removal with selective noise removal. It removes perturbations in regions in which the source model focuses while the target model does not. It would be more convincing if noises in regions that both source and target models neglect were removed.

5. In the experiments, MI and other transfer-based attacks are used as baselines but PGD attack is not presented.

6. In terms of dataset, the expression is not rigorous (the chosen images can be almost classified correctly by all models, i.e., the model has a high accuracy). Images correctly classified by all used models in this paper can be selected from ImageNet.

7. The proposed APD method can be more intuitive if the flow chart is provided.

8. The evaluation of APD’s performance is limited and not comprehensive due to a lack of diversity of model architectures. Only transferability across CNNs is presented in the paper but other popular models like ViTs and SNNs are not considered, which is important in terms of transferability evaluation. Such evaluations are mentioned in recent works like [2] [3]

[1] Xinquan Chen, Xitong Gao (equal)*, Juanjuan Zhao, Kejiang Ye, Chengzhong Xu. AdvDiffuser: Natural Adversarial Example Synthesis with Diffusion Models. International Conference on Computer Vision (ICCV). 2023.
[2] Muzammal Naseer, Kanchana Ranasinghe, Salman Khan, Fahad Shahbaz Khan, and Fatih Porikli. On improving adversarial transferability of vision transformers. arXiv preprint arXiv:2106.04169, 2021.
[3] Wenqian Yu, Jindong Gu, Zhijiang Li, Philip Torr.Reliable Evaluation of Adversarial Transferability. arXivpreprint arXiv:2306.08565, 2023

**Questions:**

1. The paper uses hotspots(i.e. the local maximum points) of the CAM as the midpoints of the dropped regions. Why not use thresholds to determine the dropped regions?

2. Will the attention regions shift during the attack iterations? Can the attention shifts be visualized?

3. The performance of APD on two classic defense methods is presented. What is the attack performance on adversarial training models?

---

> ### Author Response · Authors · 2023-11-16
>
> ## 1. Thoughts is similar to "AdvDiffuser: Natural Adversarial Example Synthesis with Diffusion Models. International Conference on Computer Vision (ICCV). 2023".
>
> **Introduction of AdvDiffuser's Method**
>
> AdvDiffuser proposes a method that utilizes Diffusion Models to generate natural adversarial examples. In Section 3.2 of their paper, they introduce the concept of **Adversarial Inpainting**, which utilizes Class Activation Maps (CAM) to derive a mask matrix. The primary goal is to ensure that the generated adversarial example closely resembles the reference image by minimally perturbing areas containing important objects, while mainly perturbing less critical areas.
>
> **Similarities with Our Work**
>
> The similarity between AdvDiffuser's approach and ours lies in both methods employing CAM for masking. However, it's crucial to note that the CAM-based mask method is a widely used technique in deep learning, especially in semantic segmentation, serving as a fundamental method. The technical novelty of both our method and AdvDiffuser's do not lie in the CAM-based mask itself.
>
> **Novelty of Our Method**
>
> The innovation of our method is twofold:
>
> 1. We discovered that a complete attention region comprises multiple blocks.
> 2. We indicated substantial synergy between these distinct blocks, as illustrated in Section 3.3 of our paper.
>
> As a result, we utilized CAM to identify positions of multiple important regions, effectively decoupling noise from these blocks.
>
> Similarly, the CAM-based mask method is not the sole innovation of AdvDiffuser's method. Their novelty lies in reducing noise in important areas to achieve the appearance of a reference image. CAM serves as their tool for identifying important objects’ positions. Moreover, the difference lies in their method normalizing the mask values to [0, 1], reducing the noise weight in important objects’ positions and increasing the noise weight in non-important positions. Moreover, their method does not employ a dropout mechanism, unlike ours, and thus, they don't achieve the decoupling effect that our method accomplishes.
>
> ## 2. About Attention Regions Shift During Attack Iterations
>
> Yes, the attention regions will shift during the attack iterations. Actually, our paper contains relevant content. The shift in attention regions is not a drawback. In fact, we utilize this shift to enhance our method. In section 3.4 of our paper, we have mention that: "It’s worth noting that we use CAMs at each attack iteration instead of just the initial image’s CAM because the attention region expands over the attack steps. The initial image’s CAM provides limited attention blocks. Using CAMs at each step allows us to identify as many attention blocks as possible over the complete attack sequence, providing a more comprehensive set of regions to drop for better transferability."
>
> About "it is unreliable to rely on the CAM from the previous iteration as guidance for the current one": Actually, we update the adversarial examples based on the CAM of the current iteration. And the shifting attention doesn’t make our method unreliable. We only aim to discover additional blocks to enable the generated adversarial samples to transfer across more models.
>
> We will incorporate visualizations illustrating the attention region shifts in the Appendix section of our paper. We appreciate your comment to enhance the clarity of our work. For now, the visualizations can be seen in "attention_region_shift.pdf" in Supplementary Material.

---

> ### Author Response · Authors · 2023-11-16
>
> ## 3. Comparisons with more relevant methods
>
> Thank you for your valuable suggestion. We have incorporated additional comparisons with advanced methods, namely PI-FGSM[1], RAP[2], SSA[3], PGN[4], TAIG[5], and CFM[6]. Partial results are presented in **Table[3]**.
>
> We appreciate your feedback and believe these enhancements strengthen the comprehensiveness of our comparative analysis.
>
> [1] PI-FGSM: Patch-wise Attack for Fooling Deep Neural Network. ECCV 2020.
>
> [2] Boosting the transferability of adversarial attacks with reverse adversarial perturbation. NeurIPS 2022.
>
> [3] Frequency Domain Model Augmentation for Adversarial Attack. ECCV 2022.
>
> [4] Boosting Adversarial Transferability by Achieving Flat Local Maxima. NeurIPS 2023.
>
> [5] Transferable Adversarial Attack based on Integrated Gradients. ICLR 2022.
>
> [6] Introducing Competition to Boost the Transferability of Targeted Adversarial Examples through Clean Feature Mixup. CVPR 2023.
>
> ### Table[3]: Attack Success Rates (\%) of adversarial attacks. We integrate our method into all the baselines, such as we let 'PI/+APD' denote PI and APD-PI. The best results are bold. * indicates the white-box setting.
>
> Note that for all methods, we integrate momentum because we take momentum as a fundamental method.
>
> #### Crafted on Inc-v3
>
> | Attack   | Inc-v3*          | Inc-v4           | IncRes-v2        | Res-101          | Inc-v3$_{ens3}$  | Inc-v3$_{ens4}$  | IncRes-v2$_{ens}$ |
> |----------|------------------|------------------|------------------|------------------|------------------|------------------|------------------|
> | PI/+APD  | **100.0/100.0**  | 56.3/**63.2**    | 51.6/**58.6**    | 45.2/**51.1**    | 21.5/**24.0**    | 20.1/**23.3**    | 11.5/**13.3**    |
> | RAP/+APD | **100.0/100.0**  | 50.1/**63.8**    | 48.8/**63.9**    | 39.2/**52.8**    | 15.1/**17.7**    | 13.7/**18.5**    | 6.4/**9.0**      |
> | SSA/+APD | **99.6/99.6**    | 54.9/**66.3**    | 53.7/**65.2**    | 48.9/**59.7**    | 15.3/**21.4**    | 15.7/**21.2**    | 6.7/**10.8**     |
> | TAIG-S/+APD | **100.0/100.0** | 75.6/**86.1**    | 75.0/**85.5**    | 66.6/**80.0**    | 22.7/**34.1**    | 21.9/**32.9**    | 12.6/**17.4**    |
> | PGN/+APD | 98.7/**98.9**    | 63.3/**73.4**    | 59.2/**69.6**    | 53.4/**63.8**    | 20.4/**27.4**    | 20.7/**25.9**    | 8.9/**11.9**     |
> | CFM/+APD | **100.0/100.0**  | 34.6/**49.7**    | 28.1/**44.3**    | 69.7/**78.9**    | 17.0/**21.0**    | 15.0/**19.7**    | 10.7/**13.2**    |
>
>
> #### Crafted on Inc-v4
>
> | Attack   | Inc-v3 | Inc-v4* | IncRes-v2 | Res-101 | Inc-v3$_{ens3}$ | Inc-v3$_{ens4}$ | IncRes-v2$_{ens}$ |
> |----------|--------|---------|-----------|---------|------------------|------------------|-------------------|
> | PI/+APD  | 68.3/**73.5** | **100.0/100.0** | 55.2/**61.3** | 47.6/**54.8** | 23.3/**26.5** | 22.3/**25.5** | 15.2/**16.2** |
> | RAP/+APD | 65.1/**79.9** | **100.0/100.0** | 51.1/**67.3** | 45.8/**60.6** | 15.1/**21.9** | 14.2/**18.6** | 6.6/**10.2** |
> | SSA/+APD | 73.9/**80.9** | 98.6/**99.0** | 60.2/**70.8** | 58.4/**68.9** | 19.1/**28.5** | 17.6/**25.5** | 8.9/**14.4** |
> | TAIG-S/+APD | 80.9/**91.7** | **100.0**/99.9 | 72.4/**87.2** | 67.4/**81.2** | 23.5/**37.9** | 21.7/**34.0** | 13.1/**21.2** |
> | PGN/+APD | 81.6/**87.0** | **99.6**/98.9 | 68.5/**80.6** | 64.6/**74.3** | 25.8/**35.1** | 25.3/**32.1** | 13.3/**17.4** |
> | CFM/+APD | 76.9/**88.1** | **100.0/100.0** | 32.6/**56.1** | 72.4/**82.8** | 17.9/**28.0** | 16.9/**25.5** | 11.7/**17.7** |
>
> #### Crafted on IncRes-v2
>
> | Attack   | Inc-v3          | Inc-v4          | IncRes-v2*      | Res-101         | Inc-v3$_{ens3}$ | Inc-v3$_{ens4}$ | IncRes-v2$_{ens}$ |
> |----------|-----------------|-----------------|-----------------|-----------------|-----------------|-----------------|-------------------|
> | PI/+APD  | 69.9/**78.7**   | 65.6/**72.6**   | **100.0/100.0** | 51.8/**63.2**   | 20.8/**25.3**   | 17.7/**23.7**   | 12.1/**15.1**    |
> | RAP/+APD | 66.4/**80.1**   | 56.9/**72.9**   | **100.0**/99.8  | 45.3/**62.3**   | 15.6/**24.6**   | 13.2/**21.2**   | 7.0/**13.7**     |
> | SSA/+APD | 82.7/**88.0**   | 71.3/**80.2**   | 98.9/**99.1**   | 67.1/**74.6**   | 23.1/**34.5**   | 20.4/**29.4**   | 10.8/**17.2**    |
> | TAIG-S/+APD | 89.0/**94.6**   | 85.5/**91.7**   | **100.0/100.0** | 78.5/**88.0**   | 28.5/**48.2**   | 25.8/**43.1**   | 17.3/**32.3**    |
> | PGN/+APD | 87.2/**90.7**   | 80.0/**83.5**   | **99.1**/98.7   | 73.1/**80.0**   | 31.6/**43.6**   | 28.3/**37.2**   | 15.7/**23.5**    |
> | CFM/+APD | 84.9/**92.0**   | 61.8/**78.1**   | **100.0/100.0** | 80.3/**86.9**   | 24.4/**39.3**   | 21.7/**34.1**   | 16.4/**27.6**    |、

---

> > ### Author Response · Authors · 2023-11-16
> >
> > #### Crafted on Res-101
> >
> > | Attack   | Inc-v3 | Inc-v4 | IncRes-v2 | Res-101* | Inc-v3$_{ens3}$ | Inc-v3$_{ens4}$ | IncRes-v2$_{ens}$ |
> > |----------|--------|--------|-----------|----------|------------------|------------------|-------------------|
> > | PI/+APD  | 70.2/**76.7** | 63.6/**68.7** | 61.5/**69.7** | **100.0/100.0** | 26.7/**32.4** | 25.3/**30.3** | 14.5/**18.4** |
> > | RAP/+APD | 68.4/**78.6** | 62.8/**69.7** | 59.0/**69.4** | **99.6/99.6** | 20.6/**27.0** | 16.9/**24.0** | 8.5/**12.4** |
> > | SSA/+APD | 80.0/**81.3** | 69.1/**72.8** | 68.5/**71.6** | **99.6/99.6** | 25.8/**33.8** | 22.4/**29.7** | 11.9/**15.1** |
> > | TAIG-S/+APD | 79.7/**84.9** | 74.1/**81.5** | 72.8/**79.6** | **100.0/100.0** | 30.1/**39.8** | 26.3/**35.4** | 16.1/**22.9** |
> > | PGN/+APD | 84.5/**84.8** | 76.1/**77.9** | 77.5/**76.4** | 99.8/**100.0** | 34.7/**41.2** | 30.7/**36.9** | 17.6/**20.8** |
> > | CFM/+APD | 83.7/**87.8** | 60.2/**65.1** | 50.2/**58.4** | **100.0/100.0** | 22.4/**28.3** | 22.3/**27.8** | 14.7/**19.3** |

---

> ### Author Response · Authors · 2023-11-16
>
> ## 4. About the Regions that Both Source and Target Models Neglect
>
> We employed different drop strategies to eliminate noise, and the results are presented in **Table[4]**. 'Attention Removal:S-T' refers to removing noise in regions where the source model focuses but the target model does not. 'Random Removal' involves randomly dropping noise of equal size to 'Attention Removal:S-T'. 'Attention Removal:-S-T' indicates removing noise in regions neglected by both the source and target models, using the same size as 'Attention Removal:S-T'. It is observed that noise in regions neglected by both the source and target models has the least influence on transferability. This aligns with the ideas presented in our paper.
>
> ### Table[4]: Attack Success Rate(\%) of when inflicting different drop strategies
>
> | Source Model | Drop Region               | Inc-v3 | Inc-v4 | IncRes-v2 | Res-101 |
> |--------------|---------------------------|--------|--------|-----------|---------|
> | Inc-v3       | Attention Removal:S-T    | /      | 31.1   | 24.5      | 22.9    |
> | Inc-v3             | Random Removal            | /      | 39.6   | 30.3      | 32.1    |
> | Inc-v3             | Attention Removal:-S-T   | /      | 41.5   | 32.6      | 39.6    |
> | Inc-v4       | Attention Removal:S-T    | 41.7   | /      | 25        | 26      |
> | Inc-v4             | Random Removal            | 52.7   | /      | 37.7      | 36      |
> | Inc-v4             | Attention Removal:-S-T   | 55.9   | /      | 41.7      | 37.9    |
> | IncRes-v2    | Attention Removal:S-T    | 40.3   | 30.5   | /         | 25.3    |
> | IncRes-v2             | Random Removal            | 51.7   | 41.7   | /         | 33.4    |
> | IncRes-v2             | Attention Removal:-S-T   | 56.8   | 45.6   | /         | 38.3    |
> | Res-101      | Attention Removal:S-T    | 39.5   | 35.5   | 27        | /       |
> | Res-101             | Random Removal            | 55.9   | 48.8   | 45.3      | /       |
> | Res-101             | Attention Removal:-S-T   | 58.1   | 51.4   | 48.9      | /       |
>
>
> ## 5. About PGD Attack
>
> The selection of adversarial example generation methods in our experiments focused on several fundamental techniques, namely FGSM, I-FGSM, PGD, C&W, and DeepFool. Our proposed method is based on I-FGSM, and consequently, we considered baselines within this branch, such as MI-FGSM, while excluding other fundamental methods like FGSM, ILCM, PGD, C&W, and DeepFool. We made this choice believing that including one fundamental baseline is sufficient for our study.
>
> Moreover, although PGD attacks were not considered in a series of related works - MI, DIM, TIM, SIM, Admix - we still conducted experiments by comparing PGD with the PGD version of our method. The results, demonstrating the effectiveness of our approach, are presented in **Table[5]**.
>
>
> ### Table[5]: Attack Success Rate of PGD Version of Our Method
> | Source Model | Attack Method | Inc-v3 | Inc-v4 | IncRes-v2 | Res-101 | Inc-v3ens3 | Inc-v3ens4 | IncRes-v2ens |
> |--------------|---------------|--------|--------|-----------|---------|------------|------------|--------------|
> | Inc-v3       | PGD           | 100.00%| 51.20% | 49.20%    | 40.00%  | 15.10%     | 13.70%     | 6.40%        |
> | Inc-v3       | PGD-APD       | 100.00%| 66.20% | 64.00%    | 54.30%  | 19.60%     | 19.00%     | 9.40%        |
> | Inc-v4       | PGD           | 64.70% | 100.00%| 52.80%    | 44.80%  | 15.90%     | 14.20%     | 7.10%        |
> | Inc-v4       | PGD-APD       | 79.40% | 100.00%| 67.00%    | 60.80%  | 21.10%     | 20.90%     | 10.80%       |
> | IncRes-v2    | PGD           | 66.60% | 57.40% | 99.90%    | 46.80%  | 16.50%     | 15.00%     | 8.30%        |
> | IncRes-v2    | PGD-APD       | 80.50% | 73.80% | 99.80%    | 62.70%  | 24.90%     | 21.20%     | 14.40%       |
> | Res-101       | PGD           | 68.70% | 62.20% | 61.40%    | 99.70%  | 22.90%     | 17.80%     | 9.90%        |
> | Res-101       | PGD-APD       | 76.70% | 70.80% | 68.00%    | 99.70%  | 30.50%     | 23.90%     | 14.20%       |
>
> ## 6. About the expression of dataset.
>
> Thank you for your reminder. The expression of dataset in our paper might not has been entirely clear. The random selection mentioned in the paper was not conducted by us but by the organizers of the NIPS 2017 adversarial attack challenge. They randomly selected 1000 images from ImageNet, which later became the most widely used dataset for adversarial examples transferability. The models we selected exhibit high classification accuracy on these images.
>
> ## 7. The flow chart.
>
> See "flow_chart.pdf" in Supplementary Material.

---

> ### Author Response · Authors · 2023-11-16
>
> ## 8. Evaluation on diversity of model architectures
>
> Thank you for highlighting this concern. We appreciate your suggestion and would like to inform you that our paper already includes an evaluation of the diversity of model architectures. In Table 3 of our original paper, we considered various network architectures, including sequencer deep LSTM, ViT-B/16, and MnasNet.
>
> Upon reviewing the paper you provided, we recognized the opportunity to further enhance the diversity of our model evaluations. As a result, we have expanded our architecture diversity by incorporating additional models, including Inc-v3, Inc-v4, IncRes-v2, Res-101, ViT-B/16, TNT, Swin, PiT-B, CaiT-S/24, Visformer-S, and SNNs (Spiking Neural Networks). It's worth noting that we selected four CNN-based models (Inc-v3, Inc-v4, IncRes-v2, Res-101), six Transformer-based models (ViT-B/16, TNT, Swin, PiT-B, CaiT-S/24, Visformer-S), and introduced SNNs to broaden the spectrum of architecture diversity. Deep learning, as a broader category, encompasses artificial neural networks (ANNs). SNNs, on the other hand, represent a distinct type of neural network within the realm of artificial neural networks.
>
> To provide you with a glimpse of the results, we have included part of the findings in **Table[8]**. Rest assured, we will incorporate the complete set of results into our paper in the final paper:
>
> ### Table[8] Attack Success Rate(\%) of models of diverse architectures
> | Source Model | Attack Method | Inc-v3 | Inc-v4 | IncRes-v2 | Res-101 | ViT-B/16 | TNT | Swin | PiT-B | CaiT-S/24 | Visformer-S | SNNs |
> |--------------|---------------|--------|--------|----------|---------|----------|-----|------|-------|-----------|-------------|------|
> | Inc-v3       | MI            | 100.00 | 49.20  | 47.10    | 36.90  | 30.10    | 32.20 | 12.30 | 14.10 | 12.40     | 19.10       | 55.00 |
> | Inc-v3       | APD-MI        | 100.00 | 68.10  | 65.70    | 56.30  | 36.10    | 41.60 | 19.30 | 27.60 | 21.70     | 30.30       | 69.70 |
> | IncRes-v2    | MI            | 66.40  | 57.40  | 99.80    | 44.10  | 33.60    | 38.00 | 12.50 | 17.10 | 13.80     | 22.90       | 60.20 |
> | IncRes-v2     | APD-MI        | 81.60  | 76.00  | 99.90    | 64.70  | 40.70    | 50.40 | 25.00 | 35.50 | 29.10     | 38.00       | 75.00 |
> | ViT-B/16     | MI            | 37.80  | 30.40  | 28.60    | 33.80  | 100.00   | 46.50 | 22.90 | 23.80 | 22.70     | 27.20       | 50.90 |
> | ViT-B/16             | APD-MI        | 44.23  | 40.16  | 34.28    | 41.52  | 100.00   | 58.26 | 39.82 | 39.37 | 35.63     | 40.72       | 56.90 |
> | Swin         | MI            | 43.90  | 36.70  | 33.60    | 36.10  | 41.70    | 47.70 | 99.80 | 35.40 | 30.60     | 40.50       | 52.70 |
> |  Swin            | APD-MI        | 44.90  | 42.90  | 35.60    | 38.20  | 49.50    | 53.10 | 99.90 | 54.30 | 44.60     | 54.40       | 53.90 |
> | CaiT-S/24    | MI            | 64.40  | 57.90  | 53.30    | 55.40  | 57.30    | 89.90 | 45.50 | 61.10 | 100.00    | 68.40       | 80.80 |
> | CaiT-S/24             | APD-MI        | 67.50  | 61.20  | 59.20    | 62.60  | 64.30    | 92.70 | 66.50 | 77.20 | 100.00    | 82.30       | 83.30 |
> | SNNs         | MI            | 52.50  | 49.50  | 39.90    | 41.00  | 39.10    | 41.30 | 14.20 | 20.60 | 19.90     | 29.70       | 100.00 |
> |  SNNs            | APD-MI        | 71.30  | 67.30  | 58.70    | 60.60  | 47.80    | 59.10 | 24.60 | 38.10 | 35.50     | 49.40       | 100.00 |

---

> ### Author Response · Authors · 2023-11-16
>
> ## 9. Why do we use hotspots instead of using thresholds to determine the dropped regions.
>
> Thank you for bringing this to our attention. We appreciate your feedback, and we acknowledge that we inadvertently overlooked an important aspect. We will include the following clarification in our paper:
>
> In our motivation experiment depicted in Fig1(b)(see in our paper), we employed a threshold to identify regions where the source model focuses while the target model does not. However, in our formal method, we opted to utilize midpoints with square regions.
>
> Here's the rationale behind this choice:
>
> 1. **Search for Effective Dropout Method**: Our goal is to discover a more effective dropout method by exploring regions where the synergy is more pronounced across models.
> 2. **Observation from Experiment (Fig1(b))**(see in our paper): Through our experiments, especially in Fig1(b), we found that the "regions where the source model focuses while the target model does not" have a significant negative impact on transferability. This observation led us to conclude that differences in attention regions between the two models contribute most to the negative impact on transferability.
>
> 3. **Identification of Attention Region Differences**: Combining the findings from points 1 and 2, it became crucial to identify the differences in attention regions across different models. From our observations in Fig2, we noted that a model's attention region comprises some meaningful blocks. By decoupling perturbations in these different blocks, we can effectively improve transferability.
>
> 4. **Challenges with Threshold-Based Approach**: When using a threshold to choose dropped regions, two challenges arise: i) if the threshold is set too low, different blocks merge together, and ii) if set too high, each dropped region does not cover one entire block. Additionally, different attention regions have different values, making it challenging to determine an optimal threshold. In contrast, the hotspot method accurately identifies each block. Therefore, we opted for the hotspot midpoint approach. The use of a square region is more of a customary choice. Shapes like circles or triangles are also workable. The emphasis is on choosing the hotspot's center rather than fixating on the shape.
>
> ## 10. The attack performance on adversarial training models?
>
> Indeed we have the evaluation on adversarial training models in our paper. The $Inc-v3_{ens3}$, $Inc-v3_{ens4}$ and $IncRes-v2_{ens}$ are adversarial training models. We test the signal model attack setting on these adversarial training models in Table 1(see in our paper) and ensemble model attack setting in Table 2(see in our paper). Putting these three classic adversarial training models and normal models together is a usual practice, and we apologize the confusion it may have caused you.

---

> ### Author Response · Authors · 2023-11-22
>
> Dear Reviewer PxjZ,
>
> Thank you again for your time. As the deadline for discussion is approaching, we do wish to hear from you to see if our response resolves your concerns. We are happy to provide any additional clarifications if needed.

---

> > ### Comment · Reviewer_PxjZ · 2023-12-01
> > **Final response to rebuttal**
> >
> > Thank the authors for providing a detailed rebuttal. Most of my concerns are well addressed.
> >
> > I have read through all the comments and rebuttals. I still have the following concerns:
> > All reviewers find that the proposed method is closely related to existing literature, such as Dropout-related Transfer attacks and attention-related transfer attacks. The proposed method is not novel enough for ICLR.
> >
> > Hence, I tend to keep my original score. I suggest the authors position this paper from an understanding perspective.

---

### Official Review · Reviewer_at4m · 2023-10-31

**Soundness:** 2 fair
**Presentation:** 3 good
**Contribution:** 2 fair
**Rating:** 5
**Confidence:** 3

**Summary:**

This paper's primary objective is to enhance the transferability of adversarial examples. To achieve this, the method introduced in the paper incorporates dropout techniques within the optimization process of adversarial attacks. The specific dropout regions are determined using Class Activation Mapping (CAM). Experimental results demonstrate that this method effectively enhances the performance compared to several baseline approaches.

**Strengths:**

The experimental results exhibit strong credibility. Adversarial examples are generated based on various base models (Inception-V3, Inception-V4, Inception-ResNet-V2, ResNet-101), and the proposed method, APD, is applied to several baseline attack techniques like MI-FGSM and DIM-FGSM. The results consistently demonstrate the enhancement provided by the proposed method.

I would also like to acknowledge the value of Table 3, which assesses the effectiveness of the proposed method on diverse models, including transformer-based models.

**Weaknesses:**

1. The motivation section requires further clarification. While it is understood that different models may emphasize different attention regions, a more explicit rationale for using Class Activation Mapping (CAM) to select dropout regions is necessary. Addressing whether random dropout region selection is a viable alternative would not only provide a valuable baseline but also shed light on the fundamental choices made in this work.

2. The paper emphasizes the distinction between traditional feature-level dropout and the proposed image-level dropout method for constructing adversarial examples. It is worth exploring the potential impact of incorporating feature-level dropout into the process and conducting a more in-depth discussion of related works such as Huang et al. (2019) and Li et al. (2020). These discussions could include experimental comparisons to better highlight the advantages and trade-offs of the proposed approach in relation to these previous works.

**Questions:**

See weakness.

**Details Of Ethics Concerns:**

This paper doesn't include additional ethical issues.

---

> ### Author Response · Authors · 2023-11-15
>
> ## 1. Motivation of using Class Activation Mapping (CAM)
>
> Here's the motivation:
>
> 1. To find more effective dropout methods, we need to explore where the synergy is more pronounced across different models.
>
> 2. Through our experiments in Fig1(b)(see in our paper), we discovered that the regions where the source model focuses while the target model does not have the most substantial negative impact on transferability. This leads us to the conclusion that the differences between the attention regions of two models have the most negative impact on transferability.
>
> 3. Combining 1 and 2 above, we need to identify the difference of attention region across different models. Observing from Fig2(see in our paper), it is noted that a model's attention region consists of some meaningful blocks. Hence, if we decouple the perturbations in different blocks, we can dropout more effectively.
>
> ## 2. About Feature-level attack
>
> Firstly, we discover that the synergy among noises at the image-level affects the transferability of the model, so we propose to enhance transferability by suppressing this synergy at the image level. As a result, directly implementing feature-level dropout seemed obtrusive, so we did not adopt feature-level dropout in our paper.
>
> However, here, we are considering the preliminary design of a feature-level dropout method to improve the transferability of adversarial examples. We carefully redesign a feature-level dropout method (called FAPD). Specifically, during the iteration process, when an image generates a feature, we apply dropout with a probability of 0.1 to this feature. When we have decided to perform dropout on a feature, we replace each neuron by the clean image's feature with a probability $p$. Inspired by [1], we choose the layers to be all convolutional layers and linear layers when the output's spatial size is less than or equal to 1/16 of the original input size.
>
> [1] Introducing Competition to Boost the Transferability of Targeted Adversarial Examples through Clean Feature Mixup. ICCV 2023.
>
> Results in Table[1] show a comparison between the baseline (RDI-MI) and ours (FAPD-RDI-MI). Results in Table[2] show a comparison among different dropout probabilities $p$.
>
> ### Table[1]: Attack Success Rate(%) of feature level attack ($p=0.5$)
>
> | Source Model | Attack Method | Inc-v3 | Inc-v4 | IncRes-v2 | Res-101 |
> |--------------|---------------|--------|--------|-----------|---------|
> | Inc-v3       | RDI-MI         | 100    | 86.4   | 79.9      | 85.4    |
> | Inc-v3       | FAPD-RDI-MI    | 100    | 98.3   | 97.5      | 98      |
> | Inc-v4       | RDI-MI         | 75.9   | 100    | 70.1      | 77.8    |
> | Inc-v4       | FAPD-RDI-MI    | 98.1   | 99.9   | 96.9      | 96.6    |
> | IncRes-v2    | RDI-MI         | 77     | 81.7   | 100       | 76.8    |
> | IncRes-v2    | FAPD-RDI-MI    | 97     | 96.6   | 100       | 92.9    |
> | Res-101      | RDI-MI         | 92.9   | 92.4   | 80.2      | 100     |
> | Res-101      | FAPD-RDI-MI    | 99     | 99.3   | 96.8      | 100     |
>
> ### Table[2]: Attack Success Rate(%) of feature level attack with different dropout probability $p$
>
> | Source Model | $p$  | Inc-v3 | Inc-v4 | IncRes-v2 | Res-101 |
> |--------------|------|--------|--------|-----------|---------|
> | Inc-v3       | 0.1  | 100    | 93.5   | 89.9      | 92.2    |
> | Inc-v3       | 0.3  | 100    | 98.3   | 97.5      | 98      |
> | Inc-v3       | 0.5  | 100    | 96.9   | 96.2      | 96.6    |
> | Inc-v3       | 0.7  | 100    | 98.5   | 96.8      | 97.5    |
> | Res-101      | 0.1  | 96.1   | 95.5   | 85.7      | 100     |
> | Res-101      | 0.3  | 98.6   | 98.1   | 93.9      | 100     |
> | Res-101      | 0.5  | 99     | 99.3   | 96.8      | 100     |
> | Res-101      | 0.7  | 99     | 99     | 96.5      | 100     |

---

> ### Author Response · Authors · 2023-11-22
>
> Dear Reviewer at4m,
>
> Thank you again for your time. As the deadline for discussion is approaching, we do wish to hear from you to see if our response resolves your concerns. We are happy to provide any additional clarifications if needed.

---

### Meta-Review · Area_Chair_yXnP · 2023-12-13

**Metareview:**

This paper presents Adversarial Perturbation Dropout (APD), a novel approach to enhance adversarial transferability by using dropout on perturbations guided by Class Activation Mapping. Extensive experiments with different models were conducted to validate APD's efficacy.

Overall, the reviewers find this paper interesting to read, but, meanwhile, raise several major concerns: 1) its technical novelty is limited, given its similarity to multiple prior works; 2) the motivation is not clear; and 3) more ablations are needed (e.g., with Vision Transformers). The rebuttal is considered, but fails to adequately address these major concerns (especially regarding novelty).

The AC encourages the authors to carefully tackle these concerns and make a stronger submission next time.

**Justification For Why Not Higher Score:**

The reviewers have expressed several major (and unaddressed) concerns regarding this paper, therefore we cannot accept its current version.

**Justification For Why Not Lower Score:**

N/A

---

### Decision · Program_Chairs · 2024-01-16

Reject